# Structural basis for SHOC2 modulation of RAS signalling

Nicholas P. D. Liau[1], Matthew C. Johnson[1], Saeed Izadi[2], Luca Gerosa[3], Michal Hammel[4], John M. Bruning[5], Timothy J. Wendorff[1], Wilson Phung[6], Sarah G. Hymowitz[1,8 ✉] & Jawahar Sudhamsu[1,7 ✉]

The RAS–RAF pathway is one of the most commonly dysregulated in human cancers[1–3]. Despite decades of study, understanding of the molecular mechanisms underlying dimerization and activation[4] of the kinase RAF remains limited. Recent structures of inactive RAF monomer[5] and active RAF dimer[5–8] bound to 14-3-3[9,10] have revealed the mechanisms by which 14-3-3 stabilizes both RAF conformations via specific phosphoserine residues. Prior to RAF dimerization, the protein phosphatase 1 catalytic subunit (PP1C) must dephosphorylate the N-terminal phosphoserine (NTpS) of RAF[11] to relieve inhibition by 14-3-3, although PP1C in isolation lacks intrinsic substrate selectivity. SHOC2 is as an essential scaffolding protein that engages both PP1C and RAS to dephosphorylate RAF NTpS[11–13], but the structure of SHOC2 and the architecture of the presumptive SHOC2–PP1C–RAS complex remain unknown. Here we present a cryo-electron microscopy structure of the SHOC2–PP1C–MRAS complex to an overall resolution of 3 Å, revealing a tripartite molecular architecture in which a crescent-shaped SHOC2 acts as a cradle and brings together PP1C and MRAS. Our work demonstrates the GTP dependence of multiple RAS isoforms for complex formation, delineates the RAS-isoform preference for complex assembly, and uncovers how the SHOC2 scaffold and RAS collectively drive specificity of PP1C for RAF NTpS. Our data indicate that disease-relevant mutations affect complex assembly, reveal the simultaneous requirement of two RAS molecules for RAF activation, and establish rational avenues for discovery of new classes of inhibitors to target this pathway.

The RAS superfamily contains 36 members in humans[2] and includes three main RAS isoforms: HRAS, KRAS and NRAS (hereafter referred to collectively as H/K/NRAS), which are the most frequently mutated in human cancers[3], and the closely related MRAS. RAS proteins are GTP-dependent intracellular molecular switches that are anchored to the plasma membrane, which activate the RAF kinases through direct binding and membrane recruitment, resulting in RAF dimerization and pathway activation[4]. Oncogenic mutations also occur in RAFs, promoting RAF dimerization[14] and drive pathway activation both dependent and independent of RAS[15]. This has motivated intense efforts towards pharmacological intervention in this pathway as an anti-cancer therapeutic strategy[16]. Although these efforts have shown efficacy in the clinic, multiple resistance mechanisms that reactivate the RAS–RAF pathway have emerged[17,18] and reinforce the importance of a deeper molecular understanding of signalling through RAS and RAF.

Prior to pathway activation, RAF is trapped in a catalytically inactive conformation by 14-3-3 through its interaction with a specific NTpS (pS365 in BRAF) that is required for 14-3-3-dependent negative regulation[9,10,19], and a phosphoserine (pS) C-terminal to RAF kinase domain[5] (CTpS). Upon RAS–RAF binding and pathway activation, a 14-3-3 dimer binds to CTpS of two RAF molecules,[6,8] inducing dimerization and the active kinase conformation[4], and increasing RAF kinase activity towards the constitutively associated substrate MEK[7,20]. A crucial step in this transition of inactive RAF to active RAF is the dephosphorylation of the NTpS residue by the protein phosphatase PP1C to prevent reversion of the active dimeric RAF to the inactive monomeric RAF[21]. PP1C modulates many pathways within the cell, and specificity for its various substrates is controlled by more than 200 regulatory proteins, most of which use linear stretches of amino acids to engage PP1C[22,23]. To specifically dephosphorylate the NTpS of RAF[24–26], PP1C relies on binding to the leucine rich repeat (LRR) protein SHOC2, as well as the RAS proteins[11–13].

Whereas some reports suggest that MRAS alone can interact with SHOC2–PP1C[11,26], others implicate additional RAS isoforms[12,13]. Although RAS binding to effector proteins can depend on its GDP- or GTP-bound state, the nucleotide state of RAS in the SHOC2–PP1C–RAS complex has not been clearly defined. Consistent with a role in modulating the RAF monomer–dimer transition, SHOC2 knockdown results in decreased RAF dimerization[27]. Genetic knockout of *Shoc2* suppresses the growth of a subset of KRAS-mutant cancer cell lines and inhibits tumour growth in mouse models of KRAS-driven lung

[1]Department of Structural Biology, Genentech, South San Francisco, CA, USA. [2]Pharmaceutical Development, Genentech, South San Francisco, CA, USA. [3]Department of Bioinformatics and Computational Biology, Genentech, South San Francisco, CA, USA. [4]Physical Bioscience Division, Lawrence Berkeley National Labs, Berkeley, CA, USA. [5]Department of Biochemical and Cellular Pharmacology, Genentech, South San Francisco, CA, USA. [6]Department of Microchemistry, Proteomics and Lipidomics, Genentech, South San Francisco, CA, USA. [7]Department of Discovery Oncology, Genentech, South San Francisco, CA, USA. [8]Present address: The Column Group, San Francisco, CA, USA. ✉e-mail: sarah@thecolumngroup.com; sudhamsu.jawahar@gene.com

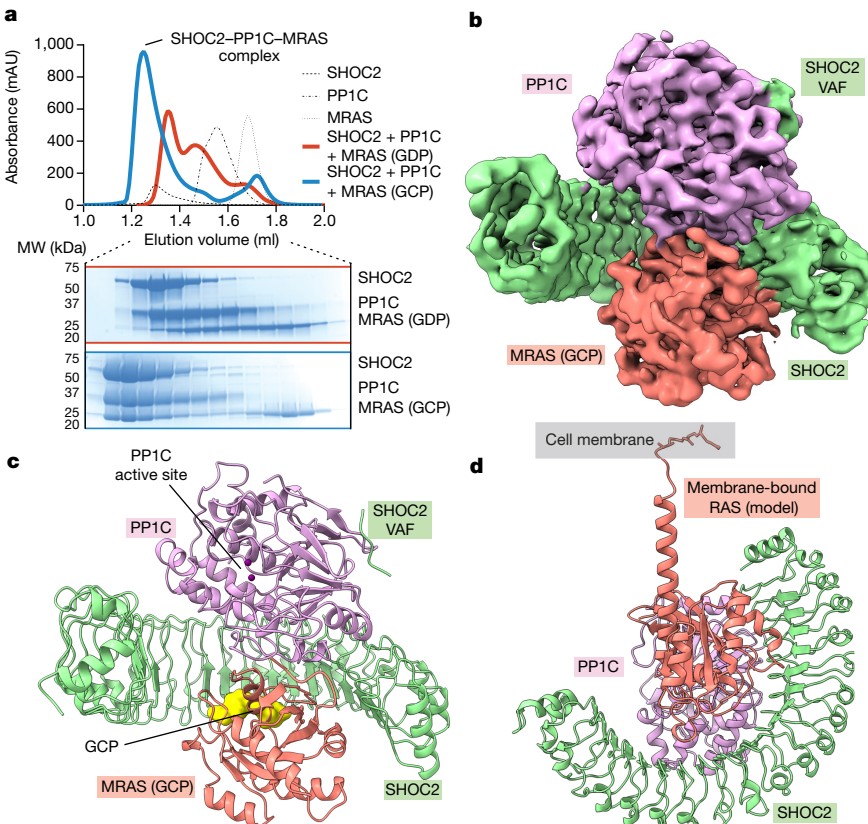

**Fig. 1 | SHOC2, PP1C and MRAS form a three-way complex in a GTP-dependent manner. a**, SEC traces showing three-way SHOC2–PP1C–MRAS complex formation in the presence of MRAS(GCP) but not in the presence of MRAS(GDP) (top), with SDS–PAGE analysis of SEC fractions (bottom). Results are representative of two independent experiments. **b**, Cryo-EM density map of SHOC2 (green) PP1C (purple) MRAS (salmon) complex, unsharpened. **c**, Structure of the SHOC2–PP1C–MRAS complex showing GCP (yellow) and the PP1C active site exposed to solvent for substrate binding. **d**, Model of the SHOC2–PP1C–RAS complex anchored to the membrane via the prenylated C terminus of RAS (salmon shows MRAS(GCP) modelled with the C-terminal helix of farnesylated KRAS; Protein Data Bank (PDB) ID: 5TAR).

cancer[28–30], demonstrating that SHOC2 function is critical for RAS–RAF pathway activation. Despite this, the structure and molecular mechanisms behind SHOC2- and RAS-driven potentiation of PP1C activity towards RAF and subsequent RAF activation have remained elusive for more than two decades. To aid in structural understanding of the mechanism of RAF activation mediated by SHOC2, RAS and PP1C, we assembled and characterized SHOC2–PP1C–RAS complexes.

## RAS binding to SHOC2–PP1C is GTP-dependent

Given the role of SHOC2 in engaging both PP1C and RAS, we reasoned that it might be possible to generate a stable complex of SHOC2, PP1C and RAS in vitro. First, we purified recombinantly expressed full-length SHOC2, the γ-isoform of PP1C (PPP1CCγ; called 'PP1C' here) and RAS proteins (HRAS, KRAS, MRAS and NRAS), loaded each RAS with either GDP or the non-hydrolyzable GTP analogue, GMP-PCP (hereafter referred to as GCP), and confirmed full nucleotide loading using mass spectrometry (Extended Data Fig. 1a). We then incubated SHOC2 with excess PP1C and RAS and assessed complex formation by size-exclusion chromatography (SEC) and found that SHOC2, PP1C and MRAS (Fig. 1a), as well as H/K/NRAS (Extended Data Fig. 1b) formed a three-way complex when RAS was GCP-bound, but not when RAS was GDP-bound, demonstrating that formation of a three-way SHOC2–PP1C–RAS complex is GTP-dependent. This selectivity for GTP-bound RAS was further confirmed using surface plasmon resonance (SPR) experiments (Extended Data Fig. 2a–c). PP1C was essential for three-way complex formation, as GCP-bound RAS was unable to form a binary

complex with SHOC2 (Extended Data Fig. 2d). In all cases when RAS was GDP-bound, we observed a SHOC2 and PP1C binary complex that did not include RAS (Fig. 1a and Extended Data Fig. 1b). MRAS(GDP) was distinct from H/K/NRAS in its ability to form a binary complex with PP1C (Fig. 1a). We sought to understand the molecular basis for this complex formation and its apparent dependence on GCP-bound RAS through biochemical and structural studies.

To confirm that we had assembled a functionally relevant SHOC2–PP1C–RAS complex, we measured dephosphorylation activity of PP1C with and without SHOC2–RAS(GCP) against a 30-residue peptide centred around the BRAF NTpS (pS365). The ternary complex exhibited higher activity against the BRAF NTpS-containing peptide substrate than PP1C alone (Extended Data Fig. 2e), suggesting that formation of the complex imparts specificity for BRAF NTpS dephosphorylation by the SHOC2–PP1C–RAS complex.

We next determined a cryo-electron microscopy (cryo-EM) structure of the 126 kDa SHOC2–PP1C–MRAS(GCP) complex to an overall resolution of 3.0 Å. The structure reveals that SHOC2 adopts a crescent-shaped architecture that is approximately 50 Å wide and 35 Å deep with 20 LRRs, and acts as a cradle to bring together PP1C and MRAS, with each protein contacting the two other proteins to form a three-way complex (Fig. 1b,c, Extended Data Fig. 3a–f and Extended Data Table 1). PP1C and MRAS, as well as the SHOC2 LRRs to which they are bound, are well resolved in the electron density map (Extended Data Figs. 3e and 4a,b). PP1C did not undergo large-scale conformational changes upon complex formation with SHOC2 compared to its individual crystal structures, whereas MRAS differed only in its dynamic

switch I and II regions (Extended Data Fig. 4c). In the cellular context, since RAS is prenylated and bound to the membrane, our structure shows how SHOC2 and PP1C would be recruited to the membrane by RAS(GTP) (Fig. 1d). No obvious membrane-interacting electrostatics were observed on the membrane-facing side of the SHOC2–PP1C–RAS complex (Extended Data Fig. 4d), suggesting that RAS is probably solely responsible for membrane recruitment of SHOC2-bound PP1C.

To further characterize SHOC2, we determined its x-ray crystal structure to a resolution of 3.2 Å (Extended Data Fig. 4e and Extended Data Table 2). Whereas the overall structure of SHOC2 alone is similar to SHOC2 in complex with PP1C–RAS, PP1C–RAS binding induces a conformational change in the SHOC2 crescent. SHOC2 undergoes a twist that is spread evenly throughout the LRRs, resulting in a tilting of the C-terminal end compared with the N-terminal end by 9° (Extended Data Fig. 4f).

We also obtained cryo-EM data and 2D class averages for a SHOC2–PP1C–KRAS complex. Preferred particle orientation precluded 3D reconstruction, although 2D class averages suggest that SHOC2–PP1C–KRAS complex adopts the same overall architecture as the SHOC2–PP1C–MRAS complex (Extended Data Fig. 3b). Molecular modelling based on small-angle x-ray scattering (SAXS) experiments further confirmed that both MRAS- and KRAS-containing ternary complexes adopt similar structures (Extended Data Fig. 5b–e). A fraction of particles in the SHOC2–PP1C–MRAS cryo-EM dataset adopted a 2:2:2 hexameric complex (a dimer of SHOC2–PP1C–MRAS heterotrimers), consistent with our SEC results (Fig. 1a and Extended Data Fig. 3b) and our SAXS data (Extended Data Fig. 5a,b,f). An apparent flexible trimer–trimer interface and preferred particle orientation prevented a high-resolution 3D reconstruction of this hexameric SHOC2–PP1C–MRAS complex.

## PP1C interaction with SHOC2

Our structure reveals that SHOC2 and PP1C interact predominantly via two large patches of charge complementarity between the two proteins (Fig. 2a). An analysis of all available structures of PP1C holoenzymes in the PDB reveals that the simultaneous engagement of these charged patches by SHOC2 represents a unique PP1C binding mode among known PP1C holoenzyme complexes. To assess the importance of these interactions for complex assembly, we measured the apparent affinity ($K_D^{app}$) of wild-type or mutated SHOC2 in forming the ternary complex using the specific dephosphorylation activity of the ternary complex for BRAF NTpS (Extended Data Fig. 6a) as a surrogate for complex formation. Charge-reversal mutations at either of these patches on SHOC2 abolished the ability of SHOC2 to form a three-way complex, whereas a charge reversal just outside the basic patch had a much less marked effect (Extended Data Fig. 6b).

PP1C-binding regulatory proteins bind to PP1C using specific small linear motifs (SLIMs), such as RVxF, SILK and KiR, for holoenzyme formation[23,31]. Analysis of all available PP1C structures revealed that PP1C uses two conserved sites on its surface to interact with regulatory subunits, with the ubiquitously engaged SLIM-binding site, and a less prominent opposing site on the diametrically opposite side of PP1C (Extended Data Fig. 6c). Motifs in the SHOC2 sequence SLVK (residues 329–332, SILK type) and KIPF (residues 369–372, RVxF type)—previously proposed to be PP1C-binding SLIMs[32]—lie on the outer surface of the SHOC2 LRR crescent and do not interact with PP1C in our structure. However, the SLIM-binding site in PP1C in our structure contains density for a bound VxF motif, probably from SHOC2 residues 64–66 (VAF) in the N terminus. High-resolution x-ray crystal structures of this region from other PP1C binding partners overlay well with this density (Extended Data Fig. 6d–f), whereas the rest of the SHOC2 N terminus (residues 1–85) was disordered (Extended Data Figs. 3e and 5d). Therefore, in addition to the predominant SHOC2–PP1C interactions via charge complementarity, our structure reveals a previously unidentified canonical PP1C binding motif in the N terminus of SHOC2.

Although SEC analysis showed that N-terminally truncated SHOC2 (SHOC2-ΔN; residues 91–582), which removes this motif (along with the disordered region in SHOC2), could form a SHOC2–PP1C–RAS complex, SHOC2-ΔN exhibited an approximately 40-fold weaker $K_d^{app}$ than full-length SHOC2 (Extended Data Fig. 7a,b), suggesting that the VAF motif in SHOC2 is functionally relevant. Our results corroborate the necessity of the N-terminal region of SHOC2 for trimer complex formation in cell-based pulldown assays[32].

Biological and binding affinity differences between the PP1C α, β and γ isoforms have been reported in the context of the SHOC2–PP1C–RAS complex, despite their high level of homology[26]. However, our structure shows that most residues that differ between the isoforms do not map to PP1C binding interfaces with SHOC2 or MRAS, and those three residues that do differ differ conservatively between isoforms (Extended Data Fig. 7c). We found that all PP1C isoforms were able to form a SHOC2–PP1C–MRAS complex by SEC (Extended Data Fig. 7d). All PP1C isoforms demonstrated the characteristic increase in catalytic activity against BRAF NTpS when part of the SHOC2–PP1C–RAS complex, and SHOC2 exhibited similar $K_d^{app}$ values for all PP1C isoforms in three-way complex formation (Extended Data Fig. 7e,f). All PP1C isoforms were also able to form a binary complex with MRAS(GDP) (Extended Data Fig. 7g). Together, these data suggest that reported differences between PP1C isoforms are caused by factors other than their inherent ability to form the SHOC2–PP1C–RAS complex, such as relative expression levels or localization within a cell.

## RAS binds to SHOC2–PP1C or RAF but not both

RAS proteins contain two dynamic regions, known as switch I and II that change conformation upon GDP–GTP exchange[33]. Our structure reveals that MRAS contacts SHOC2 primarily through these regions, providing a molecular rationale for the GTP dependence of complex formation that we observed biochemically. GTP hydrolysis to GDP is known to cause outward movement of the switch II helix. This would be expected to create a steric clash with SHOC2, as well as breaking key MRAS–SHOC2 interactions that would impair complex formation (Fig. 2b). In the GTP-bound state, key MRAS–SHOC2 interactions include E47–R177 and Q71–R288 hydrogen bonds, and an R83–R104 π-stacking interaction. A stretch of residues on switch II between Q71 and Y81 forms a network of hydrophobic interactions with SHOC2 (Fig. 2c).

A primary function of RAS in MAPK signalling is to activate RAF by binding the RAS binding domain (RBD) and cysteine-rich domain (CRD) of RAF. A comparison of the binding surface of RAS in our SHOC2–PP1C–MRAS structure to the binding surface of RAS bound to the RBD and CRD domains in CRAF[34] shows that RAS uses the same surface to engage SHOC2–PP1C and RAF(RBD–CRD), which would preclude one RAS molecule from binding RAF(RBD–CRD) and SHOC2–PP1C simultaneously (Fig. 2d). SEC analysis confirmed that neither the BRAF RBD nor the CRAF RBD could interact with SHOC2–PP1C–KRAS complex, but were able to bind free KRAS that was not part of the SHOC2–PP1C–KRAS complex (Extended Data Fig. 8a). Since binding of one RAS molecule to either SHOC2–PP1C or RAF(RBD–CRD) is mutually exclusive, these results demonstrate that two independent RAS molecules are necessary to bind SHOC2–PP1C and RAF(RBD–CRD) separately. For dephosphorylation of RAF NTpS to occur at the cell membrane, one RAS molecule would recruit SHOC2–PP1C and another RAS molecule would recruit RAF, enabling co-localization of enzyme (PP1C) and substrate (phosphorylated RAF NTpS) resulting in RAF activation.

## SHOC2–PP1C preferentially binds to MRAS

We next sought to resolve the conflicting data on whether RAS isoforms other than MRAS can bind to SHOC2–PP1C[11–13,26] in the context our own data showing that H/K/NRAS can also participate in complex formation

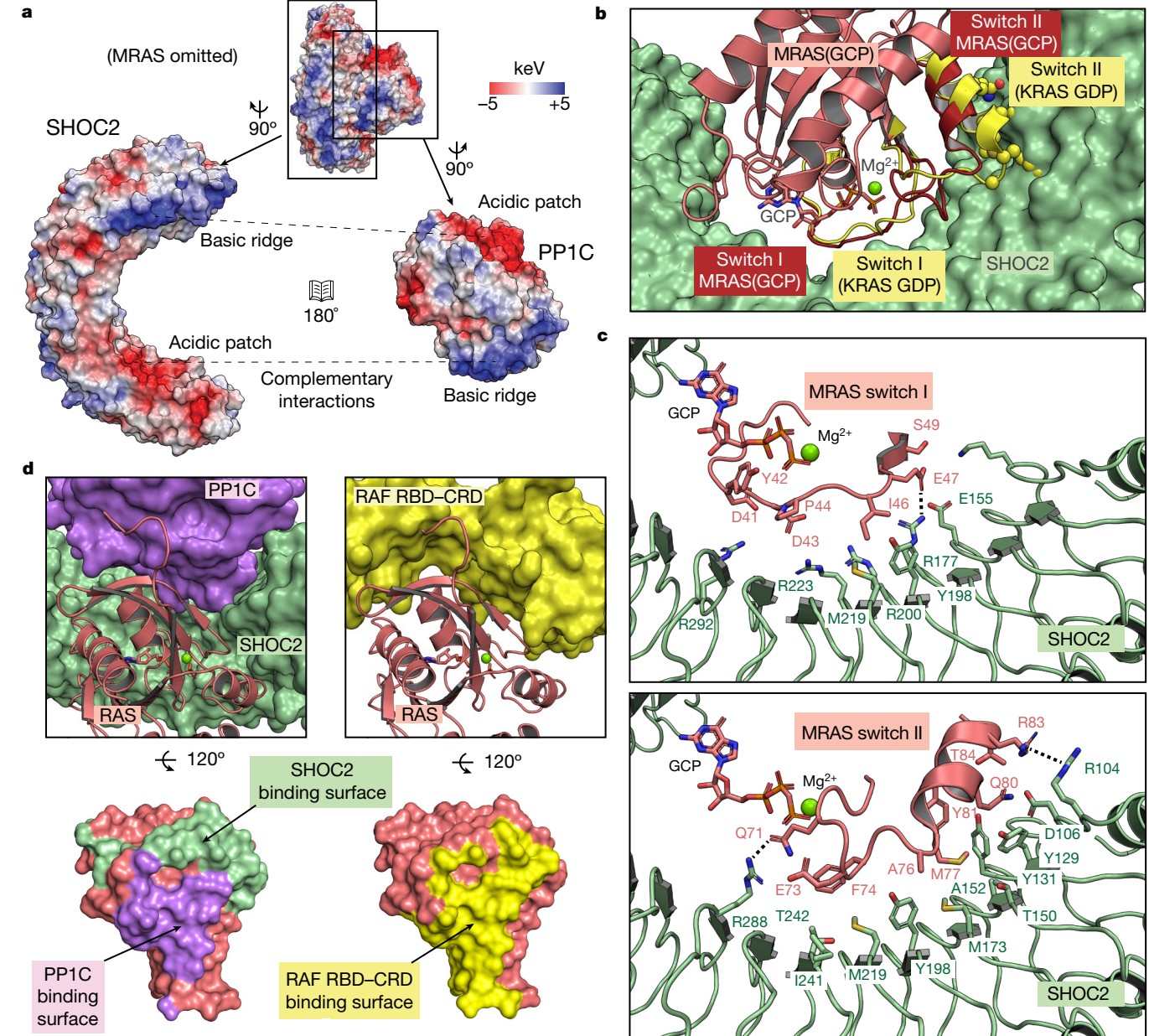

**Fig. 2 | PP1C binds SHOC2 predominantly via complementary electrostatic interactions, whereas MRAS binds SHOC2 via its switch I and II regions.**
**a**, Electrostatic surfaces of SHOC2 and PP1C create a complementary binding interface between acidic (red) and basic (blue) patches. MRAS is omitted for clarity. **b**, MRAS(GCP) (salmon) interacts with SHOC2 (green) via its switch I and II regions (dark red). Alignment of the switch I and II regions of GDP-bound KRAS (PDB: 4OBE) (yellow) showing selected sidechains (spheres), reveals

steric clashes with SHOC2. PP1C is omitted for clarity. **c**, Detailed view of MRAS(GCP) switch I (salmon, top) and switch II (salmon, bottom) interactions with SHOC2 (green), with key hydrogen bonds and π-stacking interactions shown as dashed lines. **d**, RAS (salmon) with SHOC2 (green) and PP1C (purple) binding interfaces highlighted (left), which have considerable overlap with the RAF RBD–CRD binding surface (yellow, right).

(Extended Data Fig. 2b). Since each protein in the SHOC2–PP1C–RAS complex forms direct contacts with the other two proteins, we reasoned that the binding of one partner would synergistically modulate the binding of the remaining pair[35], consistent with our dephosphorylation assay (Extended Data Fig. 8b). We measured the ability of the different RAS isoforms to increase the association of fluorescence resonance energy transfer (FRET)-labelled SHOC2 and PP1C in the context of the ternary complex. MRAS(GCP) was able to induce complex formation with a significantly lower $K_D^{app}$ (that is, stronger apparent affinity) compared with HRAS(GCP), KRAS(GCP) or NRAS(GCP) (Fig. 3a), indicating that MRAS(GCP) is the stronger binding partner

in the SHOC2–PP1C–RAS complex among the isoforms tested by a factor of 20–40-fold. H/K/NRAS still exhibited binding, albeit relatively weaker. These differences also correlated with differing abilities of KRAS and MRAS to increase PP1C–SHOC2-mediated substrate peptide dephosphorylation (Extended Data Fig. 8c).

The MRAS–PP1C interface in the ternary complex consists primarily of portions of switch I, as well as some residues N- and C-terminal to switch I (Fig. 3b). A sequence comparison of HRAS, KRAS, MRAS and NRAS also shows that some of these residues are conserved across all four of these RAS isoforms, consistent with the ability of H/K/NRAS to also form SHOC2–PP1C–RAS complexes. However, MRAS has a number

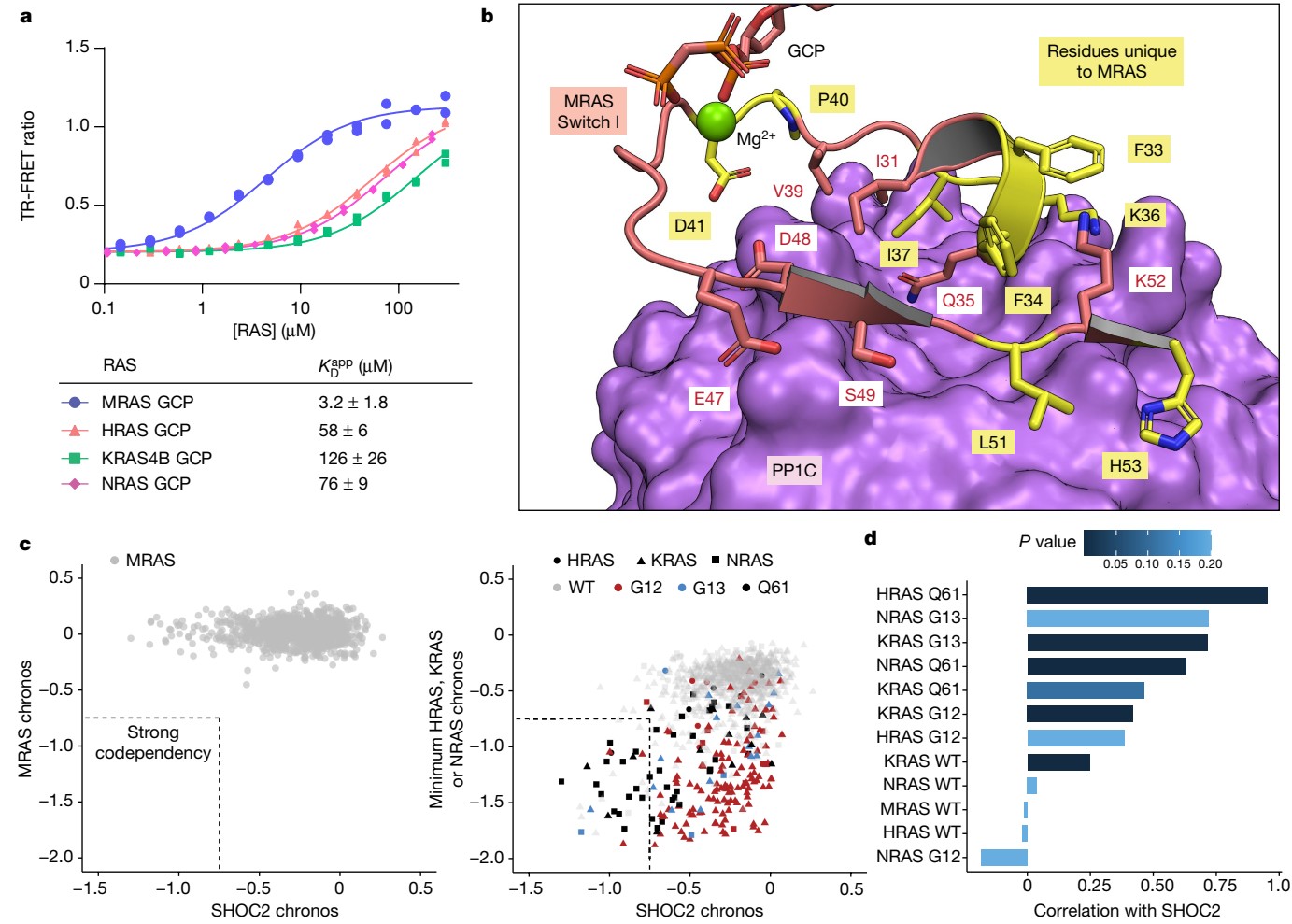

**Fig. 3 | Different RAS isoforms have a role in the SHOC2–PP1C complex.**
**a**, Top, representative result of TR-FRET measuring association between PP1C and SHOC2 in the presence of varying RAS concentrations. Data represent two time points ($n = 2$). Bottom, table summarizing $K_D^{app}$ from two independently performed experiments ($n = 2$). MRAS(GCP) is the highest-affinity binding partner of SHOC2–PP1C among those tested. $K_D^{app}$ values (shown as mean ± s.d.) are a measure of relative affinity and depend on PP1C and SHOC2 concentrations. [PP1C] = 2 nM, [SHOC2] = 200 nM. **b**, MRAS switch I PP1C-contacting regions (salmon) bound to PP1C (purple). Residues unique to MRAS versus H/K/NRAS (yellow) make substantial contacts with PP1C. SHOC2

is omitted for clarity. **c**, DepMap chronos scores for 1,061 cancer cell lines (dots) for SHOC2 versus MRAS (left) or versus the minimum chronos score among H/K/NRAS (lowest score among each of the three possible RAS isoforms for each cell line), in which cell lines are identified by the RAS isoform with minimal chronos score (shape) and the hotspot mutational status of that isoform (colour) (right). Dashed lines encapsulate cell lines with strong co-dependency (chronos scores < −0.75). **d**, Pearson correlation coefficient (bars) and *P*-value from two-sided *t*-test (colour) calculated using DepMap chronos scores for SHOC2 versus HRAS, KRAS, NRAS or MRAS in which cell lines ($n = 1,061$) were grouped by hotspot mutational status of Ras isoforms.

of unique residues in these regions, which we hypothesized might be responsible for its stronger affinity for the complex (Extended Data Fig. 8d). In addition to the switch I and switch II regions, the N- and C-terminal segments in MRAS are distinct (Extended Data Fig. 8d), and limited EM density was visible for the MRAS N-terminal tail making Van der Waals contacts with PP1C (Fig. 1b,c).

We used our time-resolved FRET (TR-FRET) binding assay to determine which of these regions contribute to the increased apparent affinity of MRAS for SHOC2–PP1C in the ternary complex. Mutation of switch I PP1C contact residues in MRAS to those found in KRAS significantly reduced MRAS binding affinity compared with wild-type MRAS (68 μM versus 3 μM, $P = 0.0004$) (Extended Data Fig. 8d), highlighting the role of the MRAS–PP1C interface in complex formation. By contrast, chimeras of KRAS with MRAS N- and/or C-terminal segments did not exhibit significantly improved affinity over wild-type KRAS. Together, these data suggest that that the switch I and surrounding residues are the primary determinants of stronger MRAS binding to SHOC2–PP1C.

## SHOC2-dependent tumours depend on H/K/NRAS

Given the ability of H/K/NRAS to form productive complexes with SHOC2–PP1C (Extended Data Fig. 1b) and the fact that they are frequently mutated in cancers, we explored the co-requirement of RAS and SHOC2 for cell fitness in an extensive panel of tumour cell lines. The Cancer Dependency Map project (https://depmap.org/portal/) provides genome-wide dependency scores for cancer cell lines using CRISPR knockout-based loss-of-function screening[36]. Dependency of a tumour cell line on a specific gene is quantified by the chronos score, which quantifies the effect of knocking out a gene on cell fitness compared with the original tumour cell. To assess the co-dependency of tumour cell lines on SHOC2 and RAS, we correlated chronos scores for SHOC2 and either MRAS or HRAS, KRAS or NRAS from all 1,061 tumour cell lines in DepMap. Despite MRAS being the stronger binding partner for SHOC2–PP1C based on our biochemical analysis, we found no correlation between SHOC2 and MRAS dependency in the tumour cell lines, demonstrating that even in the cell lines that are highly SHOC2-dependent, MRAS is dispensable (Fig 3c,d).

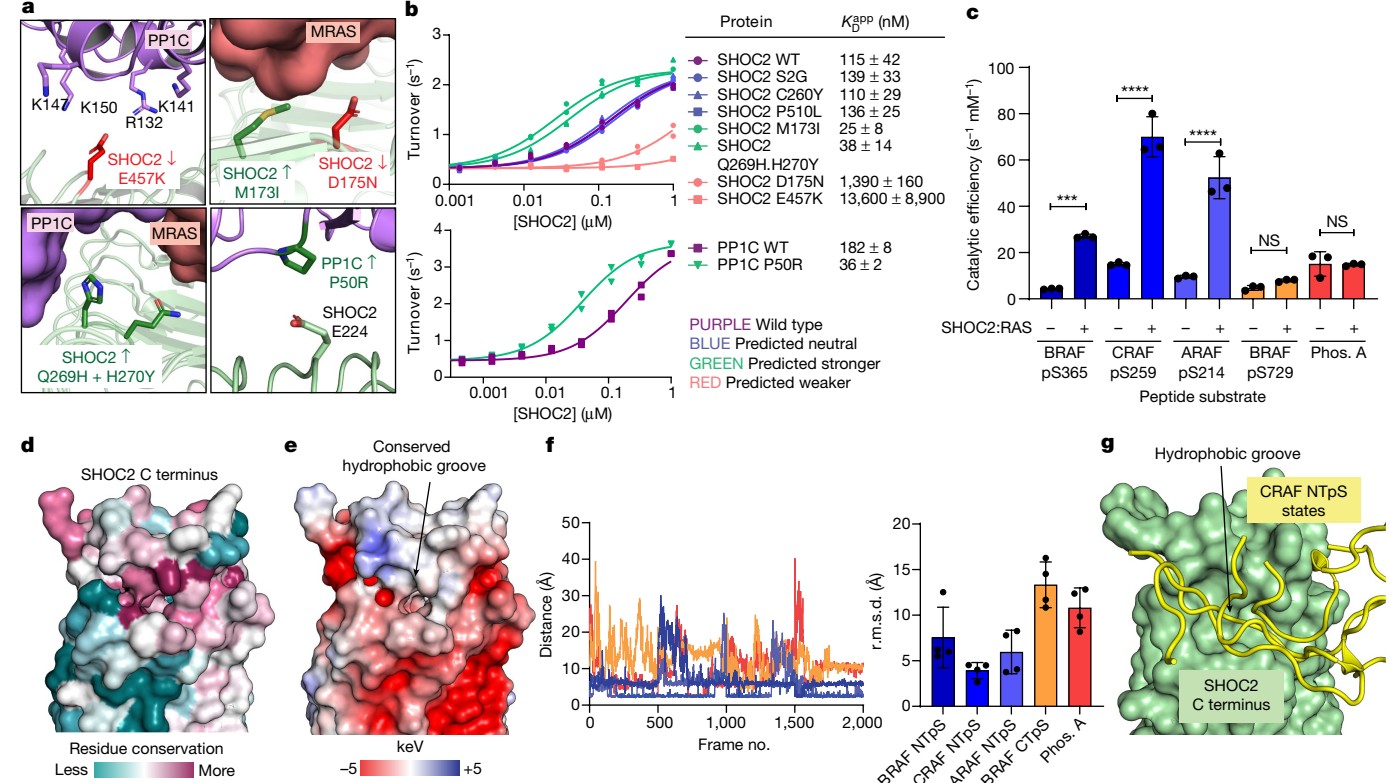

**Fig. 4 | SHOC2–PP1C–RAS substrate recognition and biological mutations.**
**a**, Mutations showing gain of function (SHOC2 M173I (top right),
SHOC2(Q269H/H270Y) (bottom left) and PP1C(P50R) (bottom right)) or loss of
function (SHOC2(E457K) (top left) and SHOC2(D175N) (top right)) map to
interfaces in the ternary complex. **b**, Representative plot of SHOC2 titrations
for SHOC2 mutants (top) or PP1C mutant (bottom) showing $K_D^{app}$ of SHOC2.
[PP1C] = 25 nM, [MRAS(GCP)] = 1 µM. Data represent peptide substrate
dephosphorylation at 2 min and 4 min time points ($n = 2$). Right, summary table
showing SHOC2 $K_D^{app}$ from three independent experiments ($n = 3$). WT, wild
type. **c**, Catalytic efficiency of peptide dephosphorylation by PP1C or SHOC2–
PP1C–RAS. Catalytic efficiency is increased when SHOC2–RAS is present. Data
are mean ± s.d. ($n = 3$). $P$ values calculated from Tukey's multiple comparison

test (one sided). ***$P = 0.001$, ****$P \le 0.001$. NS, not significant. BRAF pS729:
$P = 0.9942$, phosphorylase A pS15 (Phos. A): $P > 0.9999$. **d,e**, The C terminus of
SHOC2, showing residue conservation (**d**) and electrostatic potential (**e**).
**f**, Quantification of the pS+13 (pS+12 in BRAF) (peptide hydrophobic residues)
to Ile669 (in the SHOC2 hydrophobic groove) distance over the course of four
GaMD simulations (left), with average root mean squared deviation (r.m.s.d.)
from each of four independent simulations ($n = 4$) (right). Data are
mean ± s.e.m. **g**, Four stable states of CRAF NTpS (yellow) from PCA peaks
(Extended Data Fig. 10a), showing interaction of the peptide with the SHOC2
hydrophobic groove. PP1C and peptide N termini are omitted for clarity.
Models are aligned by SHOC2 C terminus, only one SHOC2 molecule is shown
for clarity.

By contrast, the SHOC2 chronos score significantly correlated with
H/K/NRAS chronos scores in tumour cell lines harbouring hotspot RAS
mutations at Q61, G13 and—to a smaller extent—G12 positions. For
cell lines with a high co-dependency for SHOC2 and either one of HRAS, KRAS
or NRAS (chronos score <−0.75), 71% harboured Q61, G13 or G12 hotspot
mutations (24 out of 34 cell lines) (Fig. 3c, Extended Data Fig. 8f and
Supplementary Table 1), indicating that the tumour cell growth is driven
by mutant HRAS, KRAS or NRAS, and SHOC2 knockout can phenocopy
loss of the mutant RAS in these cell lines. Our analysis also indicates
that almost all cell lines that are highly dependent on SHOC2 are also
dependent on H/K/NRAS for cell fitness. This shows that fitness defects
caused by SHOC2 inactivation in cancer cell lines are directly associated
with the oncogenic activity of H/K/NRAS and are independent of MRAS.

## Disease mutations affect complex assembly

In addition to the role of SHOC2 in tumour settings, gain- and
loss-of-function mutations in SHOC2, PP1C and MRAS have been iden-
tified in RASopathies with dysregulated MAPK signalling. Mutations
mapping to protein–protein interfaces in the ternary complex that
impair complex formation would be predicted to be less effective in
RAF activation, whereas the opposite would be predicted for mutations
that stabilize complex formation. We expressed and purified a panel of

reported SHOC2 mutant proteins and assessed the $K_D^{app}$ of each in induc-
ing increased SHOC2–PP1C phosphatase activity. Interface mutations
implicated in loss of MAPK signalling (D175N and E457K (which maps to
one of the critical SHOC2 charged patches))[37,38] showed poorer complex
association, whereas interface mutations showing a gain in MAPK signal-
ling[39,40] (M173I and Q269H/H270Y) showed stronger complex association.
A set of SHOC2 mutations that affect MAPK signalling phenotypically but
do not map to SHOC2–PP1C–RAS binding interfaces[13,41] (S2G, C260Y and
P510L) had no effect on $K_D^{app}$, suggesting that these mutations function
through means other than by affecting complex formation (Fig. 4a,b).

Aside from SHOC2, a PP1Cβ P49R mutation (P50 on PP1Cγ) has also
been identified as causing a Noonan syndrome (a RASopathy)-like pheno-
type[42,43]. $K_D^{app}$ was stronger for the P50R mutation than for wild-type PP1C.
This is rationalized structurally, as the P50R mutation would introduce
a new stabilizing ionic interaction with the neighbouring SHOC2 E224
(Fig. 4a,b). Finally, mutations in MRAS that cause Noonan syndrome[44],
such as G23V (equivalent to KRAS G13V) and T68I (equivalent to KRAS
T58I) appear in regions that do not interact directly with the ternary
complex, but which are expected to affect complex formation by increas-
ing the proportion of GTP-bound MRAS—analogous to the well-known
oncogenic KRAS mutations. Together, these results indicate that even
relatively modest fourfold to fivefold changes in affinity caused by some
of these mutations can modulate MAPK signalling sufficiently to cause

disease, and that altered SHOC2–PP1C–RAS complex association is the biochemical cause of a substantial portion of identified RASopathies.

## SHOC2–RAS drives PP1C specificity for RAF

Given that the RAF NTpS is the biological target of SHOC2–PP1C–RAS, we next sought to understand how the SHOC2–PP1C–RAS complex determines specificity for dephosphorylation of its target. As a surrogate for full-length inactive RAF, we assessed the ability of PP1C alone or a SHOC2–PP1C–KRAS complex to dephosphorylate a panel of phosphorylated peptides, enabling us to determine the catalytic efficiency ($V_{max}/K_M$; where $V_{max}$ is the velocity of the enzyme-catalysed reaction at infinite concentration of substrate and $K_M$ is the Michaelis–Menten constant) of PP1C, a measure of substrate preference. NTpS from ARAF, BRAF and CRAF (hereafter referred to collectively as A/B/CRAF), the biological targets of the SHOC2–PP1C–RAS complex, were dephosphorylated with a significantly higher catalytic efficiency by the complex compared with by PP1C alone (Fig. 4c and Extended Data Fig. 9a), indicating that the formation of the ternary complex contributes directly to the ability of PP1C to recognize the RAF NTpS substrate. Conversely, peptides that are not expected to be specific biological targets of the complex (the BRAF CTpS and the unrelated PP1C target phosphorylase A pS15 (the 'non-target peptides')) (Fig. 4c and Extended Data Fig. 9a), as well as the generic phosphatase substrate pNPP (Extended Data Fig. 9b), were dephosphorylated by PP1C with similar catalytic efficiencies, whether or not SHOC2 and RAS were present, showing that the peptide recognition by SHOC2–RAS is specific for RAF NTpS.

To test whether conserved regions in RAF beyond the NTpS also contribute to the specificity, we tested binding between the BRAF kinase domain and SHOC2–PP1C–RAS complex by SEC and did not observe complex formation (Extended Data Fig. 9c). This result, together with our previous observation that the RAF RBD does not bind to the SHOC2–PP1C–RAS complex (Extended Data Fig. 8a), suggests that the ternary complex does not bind to the ordered domains of RAF, and that the determinant of specificity is the sequence surrounding the RAF NTpS.

To better understand the structural basis for the recognition of RAF regions by SHOC2–PP1C–RAS given the weak substrate $K_M$ for the NTpS we generated a model for RAF NTpS binding to the complex and characterized it using molecular dynamics simulations. A sequence alignment of all RAF isoforms from multiple species in the NTpS region revealed that the C-terminal halves of A/B/CRAF NTpS peptides were conserved and contained several hydrophobic residues (Extended Data Fig. 9d,e). The phosphate in the pS from the NTpS must interact with the PP1C active site in SHOC2–PP1C–RAS. A surface residue conservation analysis across species for SHOC2, PP1C and MRAS revealed a conserved hydrophobic groove at the SHOC2 C terminus, as well as the previously identified hydrophobic groove next to the PP1C active site[23] (Fig. 4d,e and Extended Data Fig. 9f). As these regions neighbour each other in the structure of the complex, we hypothesized that the region of the NTpS immediately C-terminal to the pS may bind to the hydrophobic patch of PP1C adjacent to its active site (Extended Data Fig. 9f) and the subsequent residues may bind to the hydrophobic groove in the SHOC2 C terminus (Fig 4d,e). Consistent with such a model, our catalytic data suggest that outside of PP1C, specificity for substrate peptide is primarily dependent on SHOC2 rather than RAS (Extended Data Fig. 8b). SHOC2 alone can induce higher levels of PP1C activity, but RAS is able to do so only at very high concentrations (over 100 μM) (both are still synergistic because of increased total complex affinity). We therefore built models of A/B/CRAF NTpS peptides extending from the PP1C active site and hydrophobic groove into the SHOC2 hydrophobic groove, as well as models of the non-target peptides, based on the same spatial positions as the A/B/CRAF NTpS models (differing only in the peptide sequence).

To test these models, we conducted Gaussian accelerated molecular dynamics[45] (GaMD) simulations on each of these complexes to determine stable peptide–protein interactions. In all cases, the pS residue

interacted stably with the PP1C active site, as expected (Supplementary Videos 1–5). Principal component analysis (PCA) of each of the trajectories revealed several clusters of stable states for each peptide-bound complex (Extended Data Fig. 10a). For A/B/CRAF NTpS, representative structures from each stable state showed the hydrophobic C-terminal P-V/M motif interacting with the SHOC2 hydrophobic groove (Fig. 4g), whereas the same was not true of this position on the non-target peptides (Extended Data Fig. 10a). To quantify this interaction, we chose a representative residue pair from this region (peptide residue pS+13 and SHOC2 Ile669), and found that the average pairwise distance was smaller over the course of the simulations for A/B/CRAF NTpS peptides than for the non-target peptides (Fig. 4f). Together, these results suggest that hydrophobic residues on the C-terminal portion of A/B/CRAF NTpS interact with SHOC2, providing a mechanism for SHOC2–PP1C–RAS substrate recognition (Fig. 4g).

## Discussion

Whether MRAS is solely responsible for SHOC2–PP1C complex formation and RAF dephosphorylation[11,32], or whether other RAS isoforms also form functional complexes with SHOC2–PP1C[13] is a longstanding question. Similarly, it has been unclear whether GTP loading of RAS is required for productive interactions with the SHOC2–PP1C complex. This may stem from difficulties in confirming the GTP-loading status and relative concentrations of each protein in a cell. By reconstituting this system in vitro with recombinant proteins, we show that although GTP-bound MRAS is the binding partner with the highest affinity for the SHOC2–PP1C complex, GTP-bound H/K/NRAS also form productive holoenzyme complexes.

Under normal conditions, MRAS is the preferred binding partner of SHOC2–PP1C (around 20-fold higher affinity), whereas H/K/NRAS proteins are preferred binding partners of RAF[46,47] (around 10-fold higher affinity). This implies that MRAS has around 200-fold selectivity over H/K/NRAS in forming the SHOC2–PP1C–RAS complex compared with the RAS–RAF(RBD–CRD) complex, although cellular levels of each RAS protein and GTP loading status would also modulate this balance. Our analysis of the DepMap data shows that MRAS knockout is well tolerated in cancer cell lines that are strongly dependent on SHOC2 for survival, consistent with the hypothesis that H/K/NRAS may substitute for MRAS in certain contexts. This is also consistent with MRAS knockout in mice exhibiting either no phenotype or only minor defects compared with the severe consequences of H/K/NRAS knockout[48–50].

Dependency on SHOC2 was particularly evident in cancer cell lines with certain H/K/NRAS hotspot mutations (G13/Q61), indicating that SHOC2–PP1C complex formation with these mutated RAS isoforms is a likely driver of cell growth. These mutations result in aberrant RAS GTP hydrolysis and nucleotide exchange rates[51] leading to elevated RAS(GTP) compared to wild type, likely high enough to form direct complexes with SHOC2–PP1C. Consistent with MRAS being the preferential SHOC2–PP1C binding partner under basal conditions, H/K/NRAS–SHOC2 co-dependence was rarely seen in cells without hotspot RAS mutations, as high GTP-bound H/K/NRAS levels are probably not as sustained when the pathway is not constitutively active.

Growth factor induced activation of the MAPK pathway results in RAS(GTP) and recruits SHOC2–PP1C and RAF to the membrane in distinct protein complexes, since a single RAS molecule cannot simultaneously bind SHOC2–PP1C and RAF RBD–CRD. Our data show that the SHOC2–PP1C–RAS complex specifically dephosphorylates short linear RAF NTpS peptides, but does not bind to the ordered RAF domains. So, for a productive NTpS:PP1C interaction to occur, the NTpS would need to be exposed from its protective 14-3-3 binding site in the inactive RAF complex. This implies that RAS binding to RAF RBD–CRD must happen before dephosphorylation of the NTpS by PP1C bound to SHOC2–RAS. Our data also indicate that the specificity for the NTpS sequence is determined directly by PP1C and SHOC2, and that the function of RAS

in the SHOC2–PP1C–RAS complex is in the membrane localization of SHOC2 and PP1C. The simultaneous restriction of both RAS–RAF and RAS–SHOC2–PP1C complexes to the two-dimensional membrane environment would allow for co-localization of substrate RAF NTpS and phosphatase PP1C for precise spatial and temporal activation of MAPK signalling (Extended Data Fig. 10b).

Numerous cancer cell lines exhibit a strong correlation or dependency on mutant H/K/NRAS and SHOC2. Genetic knockout of SHOC2 is tolerated in adult mice[28], whereas knockout of many other components of the MAPK pathway is lethal. Further, resistance to RAS and RAF inhibition usually occurs by re-activation of the MAPK pathway. This indicates that the SHOC2–PP1C–RAS complex is an attractive therapeutic target. Our data reveal that the function of the SHOC2–PP1C–RAS complex could be targeted through a variety of mechanisms including changing the conformation of the RAS switch II, disruption of the complex by occlusion of binding interfaces, or by targeting the extended substrate binding surface extending from PP1C to SHOC2. Further inhibition of the MAPK pathway through nodes orthogonal to existing MAPK pathway inhibitor targets, such as the SHOC2–PP1C–RAS complex, may help to downregulate ERK signalling more effectively in relevant tumours.

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

## Methods

### Expression and purification of SHOC2

SHOC2 constructs (*Homo sapiens* residues 2–582, (except SHOC2ΔN, residues 91–582)) were expressed in *Spodoptera frugiperda* 9 (Sf9) cells. Cells were collected and stored at −80 °C. Cell pellets were resuspended in 100 ml per 1 l pellet of SHOC2 SEC buffer (10% glycerol, 20 mM Tris pH 7.5, 500 mM NaCl, 1 mM TCEP) supplemented with 20 mM imidazole, 1 EDTA-free protease inhibitor tablet, 0.2 mg Benzonase, and lysis detergents 0.3 % Sb3-14 and 0.03% C7BzO. The solution was homogenized and incubated with 1 ml per 1 l pellet of Ni-charged MagBeads (Genscript) for 1 h at 4 °C. Beads were washed with SHOC2 SEC buffer + 20 mM imidazole and protein was eluted with SHOC2 SEC buffer + 500 mM imidazole. SHOC2 was cleaved with TEV, dialysed into SHOC2 SEC buffer overnight at 4 °C, and re-applied to nickel beads. Flow-through containing SHOC2 was concentrated and run on a Superdex S200 16/600 in SHOC2 SEC buffer. Fractions containing purified SHOC2 were pooled, concentrated and snap frozen on liquid nitrogen.

### Expression and purification of PP1C

PP1C constructs (*H. sapiens* residues 1–323) were expressed in *Escherichia coli* BL21(DE3) Tuner cells in Terrific Broth media supplemented with 1 mM $MnCl_2$. Expression was induced at $A_{600} = 1.0$ with 50 µM IPTG and expression occurred overnight at 17 °C. Cells were collected and stored at −80 °C. Cell pellets were resuspended in 100 ml per 1 l pellet of PP1C SEC buffer (20 mM Tris pH 7.5, 700 mM NaCl, 1 mM $MnCl_2$, 1 mM TCEP) supplemented with 20 mM imidazole, 1 EDTA-free protease inhibitor tablet, 0.2 mg benzonase, lysis detergents 0.3% Sb3-14 and 0.03% C7BzO and 20 mg lysozyme. The solution was homogenized and incubated with 2 ml per 1 l pellet of Ni-charged MagBeads (Genscript) for 1 h at 4 °C. Beads were washed with PP1C SEC buffer + 20 mM imidazole and protein was eluted with PP1C SEC buffer + 500 mM imidazole. PP1C was TEV cleaved, dialysed into PP1C SEC buffer overnight at 4 °C, and re-applied to nickel beads. Flow-through containing PP1C was concentrated and run on a Superdex S75 16/600 in PP1C SEC buffer. Fractions containing pure PP1C were pooled, concentrated and snap frozen on liquid nitrogen.

### Expression and purification of RAS

H/K/NRAS constructs (*H. sapiens* residues: HRAS, 2–189; KRAS, 1–188; NRAS, 2–189; KRASΔHVR, 1–169) were expressed in *E. coli* BL21(DE3) cells in TB autoinduction medium at 17 °C for 48 h. MRAS constructs (*H.* sapiens residues 1–208) and KRAS–MRAS chimeras were expressed in Sf9 cells. Cells were collected and stored at −80 °C. Cell pellets were resuspended in 100 ml per 1 l pellet of RAS SEC buffer (20 mM Tris pH 7.5, 150 mM NaCl, 1 mM $MgCl_2$, 1 mM TCEP) supplemented with 20 mM imidazole, 1 EDTA-free protease inhibitor tablet, 0.2 mg benzonase, lysis detergents 0.3 % Sb3-14 and 0.03% C7BzO and 20 mg lysozyme (for *E. coli*-expressed proteins). The solution was homogenized and incubated with 2 ml per 1 l pellet of Ni-charged MagBeads (Genscript). Beads were washed with RAS SEC buffer + 20 mM imidazole and protein was eluted with RAS SEC buffer + 500 mM imidazole. RAS was TEV cleaved, dialysed into RAS SEC buffer + 10 µM GDP overnight at 4 °C, and re-applied to nickel beads. RAS containing flow-through was concentrated and run on a Superdex S75 16/600 in RAS SEC buffer + 10 µM GDP. Fractions containing pure GDP loaded RAS were pooled, concentrated and snap frozen on liquid nitrogen.

For GCP-loaded RAS constructs, purified RAS was mixed with a 50-fold molar excess of GCP, 10 U alkaline phosphatase agarose beads (Sigma) and incubated with agitation at 37 °C for 1 h. Protein was buffer-exchanged into 25 mM Tris pH 7.5, 100 mM NaCl, 5 mM $MgCl_2$, 10 µM GCP and snap frozen in liquid nitrogen.

### Mass spectrometry

Fifty micrograms of each sample was buffer-exchanged into 50 mM ammonium acetate, pH 7. Samples were directly infused using a TriVersa NanoMate (Advion) and analysed online via nanoelectrospray ionization with a 5 µm nozzle ESI chip (Advion) into a Thermo Exactive Plus EMR Orbitrap mass spectrometer (Thermo Fisher Scientific). Acquired mass spectral data were analysed using UniDec software[52].

### Analytical SEC

Individual proteins or complexes were run on a size-exclusion column equilibrated in 20 mM Tris pH 7.5, 100 mM NaCl, 1 mM TCEP, 0.5 mM $MnCl_2$, 0.5 mM $MgCl_2$. Fractions were analysed by SDS–PAGE stained with coomassie blue. Columns used were: Fig. 1a and Extended Data Figs. 1b, 2d, 7a,d, 8a and 9c: S200 3.2/300; Extended Data Fig. 7g: S75 3.2/300.

### Cryo-EM grid preparation and data collection

To prepare grids for electron microscopy, SEC pure SHOC2–PP1C–RAS complex was diluted to approximately 11 µM in RAS SEC buffer + 10 µM GCP, 4 µl of which was applied to holey gold grids (Ultrafoil R1.2/1.3; Quantifoil) that had been glow-discharged for 20 s using a Solarus plasma cleaner (Gatan). Grids were blotted for 3.5 s using a Vitrobot (Thermo Fisher Scientific) set to 4 °C and 100% relative humidity, and plunged into liquid ethane.

To prepare graphene oxide coated grids, the perpetually hydrated method was used[53]. Holey gold grids (Ultrafoil R1.2/1.3; Quantifoil) were glow-discharged for 20 s using a Solarus plasma cleaner (Gatan), and 4 µl graphene oxide flakes (Sigma-Aldrich) freshly diluted to 0.2 mg ml$^{-1}$ in DDI water were applied to the front of the EM grid (side with holey layer). After 50 s incubation, 4 µl of SEC pure SHOC2–PP1C–RAS complex (diluted to approximately 1 µM in RAS SEC buffer + 10 µM GCP) were applied to the back of the EM grid (side with mesh). Without further incubation, grids were blotted for 3.5 s using a Vitrobot (Thermo Fisher Scientific) set to 4 °C and 100% relative humidity, and plunged into liquid ethane.

Data collection was performed using a Titan Krios (Thermo Fisher Scientific) operating at 300 keV, equipped with a BioQuantum energy filter (20 eV slit width) and a K3 (Gatan) direct detection camera. Movies were recorded in super-resolution mode, at a nominal magnification of 105,000× (calibrated pixel size 0.419 Å), 60 frames per movie, 50 ms per frame, and electron fluence per frame of 1.07 e$^-$ Å$^{-2}$ (total fluence 64 e$^-$ Å$^{-2}$). Data collection was automated using SerialEM[54] with a set defocus range of 0.5 to 1.5 µm.

### Cryo-EM data processing and model building

Initial steps of cryo-EM data were processed using CryoSPARC Live[55]. A total 10,780 of movies were motion corrected and CTF corrected. Micrographs with a CTF fit poorer than 7.0 Å were immediately discarded, leaving 5,060 micrographs. Particles were picked using the blob picker tool and 2D classified. 2D classes representing protein complex were used to re-pick, resulting in a stack of 3,996,056 particles, which were exported to CryoSPARC, where all further processing was undertaken. Particles were subjected to 2D classification to remove junk particles, leaving 1,261,442 particles. An ab initio Reconstruction was performed with 6 3D classes, revealing 4 classes of trimeric and 1 class of hexameric SHOC2–PP1C–MRAS volumes, along with 1 class of poorly resolved junk. A total of 551,091 particles encompassing the 2 best trimeric 3D classes were combined and used for non-uniform refinement[56], with the best ab initio 3D volume as the starting model. Micrographs with a CTF fit poorer than 4.0 Å were further discarded, leaving 3,847 micrographs, and particles with a separation distance of less than 20 Å were also discarded. Remaining particles were subject to a further round of 2D classification and poorly resolved 2D classes were discarded, leaving 323,910 particles, which were re-extracted using a box size of 256 pixels. These particles were subjected to a final non-uniform refinement, including refinement of per-particle defocus and per-group CTF parameters (tilt and trefoil). This resulted in a final model with a gold standard Fourier shell correlation resolution of 2.95 Å.

Our crystal structure of SHOC2 (PDB: 7DS1), and existing structures of MRAS (PDB: 1X1S) and PP1C (PDB: 4MOV) were initially placed into the density map with ChimeraX[57]. The refinement and building were undertaken with Phenix[58] Real Space Refine and COOT[59] respectively. Molecular visualizations were created with ChimeraX and PyMol.

## SHOC2 crystal structure determination

Crystals of SHOC2 were obtained using hanging-drop vapour diffusion with 1 µl of 6.0 mg ml$^{-1}$ protein in SHOC2 SEC buffer mixed with 1 µl mother liquor (100 mM Tris pH 8.5, 200 mM MgCl$_2$, 14%(w/v) PEG4000) over a reservoir of mother liquor at 16 °C. X-ray diffraction data was collected at the Stanford Synchrotron Radiation Lightsource beamline 12-2 with an X-ray wavelength of 0.97946 Å at 100 K. Data were integrated and scaled with XDS[60]. To obtain phases, molecular replacement was attempted with various homologous LRR proteins, where PDBL 4U06 (*Leptospira interrogans* LRR protein LIC10831, 23% sequence identity) resulted in a solution. The SHOC2 chain was initially placed with Phenix Autobuild, then iteratively built in COOT[59] and refined in Phenix[58].

Towards the end of this process, the predicted structure of SHOC2 became available in the AlphaFold Protein Structure Database[61] (https://alphafold.ebi.ac.uk/entry/Q9UQ13), and the LRR portion of the predicted structure was used as a molecular replacement solution. After building and refinement, $R_{work}/R_{free}$ improved markedly to 0.209/0.238 vs 0.269/0.300 for the manually built structure. Therefore, the AlphaFold based molecular replacement solution with manual rebuilding and refinement against our experimental data resulted in the final model. Final Ramachandran statistics were 96.1% favoured, 3.9% allowed, 0% outliers.

## SEC−multi-angle light scattering−SAXS

Data were collected at the ALS beamline 12.3.1 LBNL Berkeley, California[62,63] with an X-ray wavelength of 1.127 Å. All experiments were performed at 20 °C and data was processed as previously described[64]. In brief, a SAXS flow cell was directly coupled with an HPLC system. A 55 µl volume of each sample was run through SEC and 3-s X-ray exposures were collected continuously during a 30-min elution. The SAXS frames recorded prior to the protein elution peak were used to subtract all other frames. The subtracted frames were investigated by radius of gyration ($R_g$) derived by the Guinier approximation $I(q) \approx I(0)e^{-q^2R_g^2/3}$ with the limits[65] $qR_g < 1.5$, where $q$ is the scattering vector. The elution peak was mapped by comparing the integral of ratios to background and Rg relative to the recorded frame using the program SCÅTTER. Final merged SAXS profiles, derived by integrating multiple frames at the elution peak, were used for further analysis. The program SCÅTTER was used to compute the $P(r)$ function.

## SAXS solution structure modelling

The model for full-length SHOC2 was built based on our x-ray crystal structure with the addition of missing N- and C-terminal regions in MODELLER[66]. BILBOMD[67] was used to model conformational flexibility of the N-terminal region. The experimental SAXS profiles of SHOC2 were then compared to theoretical scattering curves of the atomistic models generated by BILBOMD using FOXS[68,69], followed by multi-state model selection by MultiFoXS[70,71].

The initial atomistic models of SHOC2−PP1C, SHOC2−PP1C−MRAS and SHOC2−PP1C−KRAS were built based on our cryo-EM structure. Minimal molecular dynamics simulations were performed on flexible regions in the models by the rigid body modelling strategy BILBOMD in order to optimize conformational space of SHOC2 N-terminal region. The selection of multi-state models was performed as described above for SHOC2.

## Phosphatase assays

For peptide dephosphorylation assays, PP1C was mixed with varying concentrations BRAF pS365 peptide, SHOC2 and/or RAS at concentrations specified in each figure. At 2 min and 4 min time points, the reaction was stopped by the addition of 20 µl of reaction mix to 80 µl malachite green solution and incubated 30 min at room temperature for colour development. A standard curve of 0−100 µM inorganic phosphate was also generated. Absorbance was read at 640 nm and the enzymatic turnover for each reaction was calculated by reference to the standard curve.

For PNPP dephosphorylation assay, 10 nM PP1C, 1 µM SHOC2 and 1 µM KRAS were mixed with varying concentrations of PNPP. Absorbance was read at 405 nm at 1 min intervals. Turnover was calculated by reference to the extinction coefficient of NPP (dephosphorylated PNPP $A_{405} = 18,000$ M$^{-1}$ cm$^{-1}$) and was averaged across 5, 10, 15 and 20 min time points. Graphs were generated with GraphPad Prims8 for phosphatase assays and other biochemical assays.

## Surface plasmon resonance

Avi-tagged KRAS was expressed and purified as described above. KRAS was biotinylated with a BirA biotin-protein ligase kit (Avidity) according to the manufacturer's instructions. All SPR steps were performed on a Biacore S200 Instrument (Cytiva) at a temperature of 20 °C. In PBS a C1 chip (Cytiva) was functionalized with ~1,000−1,200 RU Neutravidin by activation with EDC/NHS followed by injection of 100 µg ml$^{-1}$ Neutravidin (prepared in 10 mM sodium acetate, pH 4.6), and capping using 1 M ethanolamine. The surface was then conditioned with 1M NaCl/50 mM NaOH. After priming the system into data collection buffer (50 mM HEPES, pH 7.4, 100 mM NaCl, 500 µM TCEP, 0.01% Tween-20, 10 mM MgCl$_2$, 500 nM GDP or GNP), biotinylated Avi-tagged KRAS pre-loaded with either GDP or GNP was then captured to ~40 RU using a brief injection of 10 s at 2 µl min$^{-1}$, and remaining biotin-binding sites were blocked with 200 nM biotin for 60 s at 100 µl min$^{-1}$. A twofold dilution series for each sample was injected sequentially from low to high concentration over the KRAS- coupled surfaces in multi-cycle kinetics mode, at a flow rate of 50 µl min$^{-1}$ for 45 s, monitoring dissociation for 300 s.

All analysis was performed using the S200 Evaluation software after applying standard double-referencing. Theoretical $R_{max}$ (the maximal feasible SPR signal generated by an interaction between a ligand−analyte pair; presented in response units (RU)) was determined as the (molecular mass of the protein in solution)/(molecular mass of the immobilized target) × (amount of immobilized target captured), and the per cent surface activity was determined as the (experimental $R_{max}$)/(theoretical $R_{max}$).

## TR-FRET binding assay

C terminally SNAP-tagged PP1C and N-terminally SNAP-tagged SHOC2 were expressed and purified as described above. PP1C was labelled with SNAP-Lumi4-Tb labelling reagent (Cisbio) and SHOC2 was labelled with SNAP-Red labelling reagent (Cisbio) as per the manufacturer's protocols. Sets of 20 µl solutions were made with final concentrations of 2 nM Tb−PP1C, 200 nM Red−SHOC2 and variable RAS concentrations, each component having been diluted with TR-FRET Assay buffer (25 mM Tris pH 7.5, 200 mM NaCl, 2 mM MgCl$_2$, 2 mM MnCl$^2$, 1 mM TCEP, 0.2 mg ml$^{-1}$ BSA) and incubated at room temperature for 30 min. TR-FRET was read with excitation at 340 nm and emission at 620 nm and 665 nm. A second reading of each plate was taken after a 10 min delay to confirm that binding had reached equilibrium. The ratio of 665 nm emission was divided by 620 nm emission at each time point to give TR-FRET ratio.

## DepMap data analysis

DepMap release Public 21Q4 datasets containing cell line information (sample_info.csv), chronos scores (CRISPR_gene_effect.csv) and mutational status (CCLE_mutations.csv), were downloaded from https://depmap.org/portal/download/. We assembled a dataset containing chronos scores for SHOC2, MRAS, KRAS and NRAS in 1,061 cell lines and annotated each cell line with the corresponding hotspot mutational

status at the position G12, G13 and Q61 for KRAS, NRAS and HRAS. Overall dependency of each cell line on H/K/NRAS was calculated as the minimal value among H/K/NRAS chronos scores. The Pearson correlation coefficient and two-sided $t$-test $P$-value between SHOC2 and MRAS, HRAS, KRAS or NRAS chronos scores, gated by mutational status on G12, G13 and Q61 positions, were calculated across the 1,061 cell lines using the cor.test function in R. Data analysis was performed using R custom scripts.

## Molecular dynamics

The complex structures were parametrized using FF19SB force field for proteins[72] and parameters for phosphorylated amino acids[73], fully solvated in OPC[74] water boxes extending 12 Å from protein edges. The GCP ligand was parameterized using the Hartree-Fock/6-31 G* basis set with Gaussian09 to calculate the restraint electrostatic potential (RESP) charges, and antechamber to RESP fit the calculated potentials to generate the force field files. Na+ counter ions were added to neutralize the system. The GPU implementation of Amber 2018 with the SPFP precision model[75] was used for the molecular dynamics simulation. First, the structure was relaxed with 2,000 steps of conjugate-gradient energy minimization in which the solutes and solvent were kept fixed. Then, the solvent molecules were allowed to move using the NPT ensemble with a temperature of 310 K. Another step of conjugate-gradient energy minimization was performed with 2,000 steps while removing all the restrains. Next, the pressure was maintained at 1 atm and the thermostat temperature increased to 310 K over the course of 500 ps, while Harmonic positional restraints of strength 10 kcal mol$^{-1}$ Å$^{-2}$ was applied to the solute. The system was then equilibrated for 1 ns with a restraint force constant of 1 kcal mol$^{-1}$ Å$^{-2}$. All restraints were removed for the production stage. The hydrogen mass repartition was used allowing a time step[76] of 4 fs. A 10 Å cut-off radius was used for range limited interactions, with particle mesh Ewald electrostatics for long-range interactions. The production simulation was carried out using NPT conditions. Langevin dynamics[77] was used to maintain the temperature at 310 K with a collision frequency of 3 ps$^{-1}$. During dynamics the SHAKE algorithm[78] was applied. Default values were used for all other simulation parameters. The production stage of the conventional molecular dynamics simulation was performed for 50 ns.

Each one of the four independent conventional molecular dynamics runs were followed by GaMD simulation[45,79] module in AMBER 2018, which included 200-ps short cMD simulation used to collect potential statistics, 2-ns equilibration after adding the boost potential, and finally, 500-ns GaMD production runs. All GaMD simulations were run at the dual-boost level by setting the reference energy to the lower bound. The average and s.d. of the system potential energies were calculated in every 50,000 (100 ps). The upper limit of the boost potential s.d., $\sigma_0$ was set to 6.0 kcal mol$^{-1}$ for both the dihedral and total potential energetic terms. The simulation frames were saved every 20 ps for analysis.

CPPTRAJ[80] was used to postprocess the ensembles from the combined GaMD trajectories. Contacts were determined by a simple distance cut-off (5 Å) between any two atoms within each residue. PCA was performed on the dihedral angles of the full structures. Each frame was RMS-fit to the first frame to remove global translational and rotational motion. The eigenvectors and eigenvalues were then obtained from diagonalization of the combined covariance matrix, after which coordinates from each independent trajectory were projected along eigenvectors of interest to obtain projection values for given modes. The isolated first principal components, Mode1 and Mode2, showing the largest variation in the data, were used to generate the 2D PCA density plots.

## Reporting summary

Further information on research design is available in the Nature Research Reporting Summary linked to this paper.

## Data availability

Coordinates and related data for structures of SHOC2 and SHOC2–PP1C–MRAS complex have been deposited in the PDB and Electron Microscopy Data Bank (EMDB), respectively, with PDB accession code 7SD1 for SHOC2 and PDB code 7SD0 and EMDB code EMD-25044 for the SHOC2–PP1C–MRAS complex. SAXS data and atomistic models have been deposited at the Small Angle Scattering Biological Data Bank (SASBDB) as SASDMB5 (SHOC2), SASDMC5 (SHOC2–PP1C), SASDMD5 (SHOC2–PP1C–KRAS) and SASDME5 (SHOC2–PP1C–MRAS). DepMap release Public 21Q4 datasets containing cell line information (sample_info.csv), chronos scores (CRISPR_gene_effect.csv) and mutational status (CCLE_mutations.csv) are publicly available from https://depmap.org/portal/download/.

## Code availability

Code used to analyse the DepMap data for SHCO2–RAS correlation will be provided upon request.

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

**Acknowledgements** We thank the BioMolecular Resources (BMR) group at Genentech for their support in generating constructs and BEVS biomass for protein production; Stanford Synchrotron Radiation Lightsource (SSRL) for synchrotron access; Advanced Light Source (ALS) for access to the SIBYLS beamline; A. Estevez for assistance with preliminary electron microscopy screening; C. Arthur for cryo-EM data collection; W. Wang for X-ray diffraction data collection; A. Oh and B. Martin for advice on Ras GTP–GDP loading; F. Shanahan and I. Yen for discussions about RAS biology; and the The 2021 CCP4/APS School in Macromolecular Crystallography for training. Use of the Stanford Synchrotron Radiation Lightsource, SLAC National Accelerator Laboratory is supported by the US Department of Energy, Office of Science, Office of Basic Energy Sciences under contract no. DE-AC02-76SF00515. The SSRL Structural Molecular Biology Program is supported by the DOE Office of Biological and Environmental Research, and by the National Institutes of Health, National Institute of General Medical Sciences (P41GM103393). The contents of this publication are solely the responsibility of the authors and do not necessarily represent the official views of NIGMS or NIH. Funding for the SIBYLS beamline at the Advanced Light Source was provided in part by the Offices of Science and Biological and Environmental Research, U.S. Department of Energy, under Contract DE-AC02-05BH11231 and NIGMS grant P30 GM124169-01, ALS-ENABLE.

**Author contributions** J.S. and S.G.H. oversaw execution of all aspects of the project. N.P.D.L. performed all of the protein biochemistry and biochemical assays (activity and TR-FRET) and solved the crystal and cryo-EM structures. M.C.J. performed cryo-grid sample preparation, collected cryo-EM data and assisted in cryo-EM data processing. S.I. performed molecular dynamics simulations and analysed data. L.G. performed bioinformatics analysis on the tumour cell line data from DepMap. M.H. performed and analysed SAXS experiments. J.M.B. performed and analysed SPR data. T.J.W. designed protein constructs and developed the TR-FRET assay. W.P. performed all mass spectrometry experiments. N.P.D.L., S.G.H. and J.S. wrote the manuscript with input from all authors.

**Competing interests** N.P.D.L., M.J., L.G., S.I., J.M.B., T.J.W., W.P. and J.S. are all employees of Genentech.

**Additional information**
**Correspondence and requests for materials** should be addressed to Sarah G. Hymowitz or Jawahar Sudhamsu.

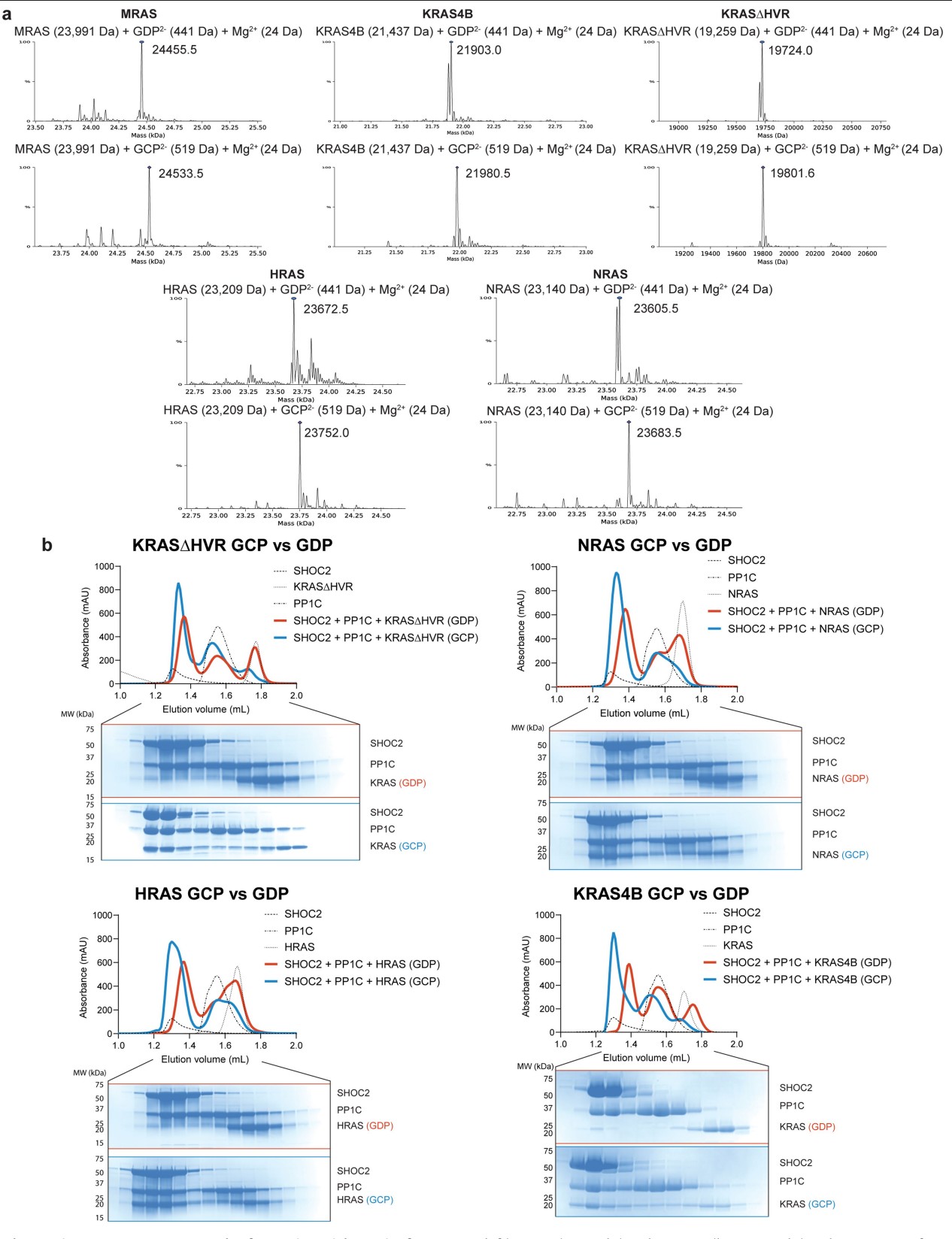

**Extended Data Fig. 1 | SHOC2:PP1C complex formation with RAS isoforms.** **a**, Nucleotide loaded RAS samples were analyzed by mass spectrometry, which found complete loading of GDP or GCP for each sample. **b**, SEC traces showing three-way complex formation between KRASΔHVR (top left), HRAS (bottom left), NRAS (top right) and KRAS4B (bottom right) in the presence of GCP loaded RAS (blue) but not GDP loaded RAS (red), and associated SDS-PAGE analysis of SEC fractions. Results representative of two independent experiments.

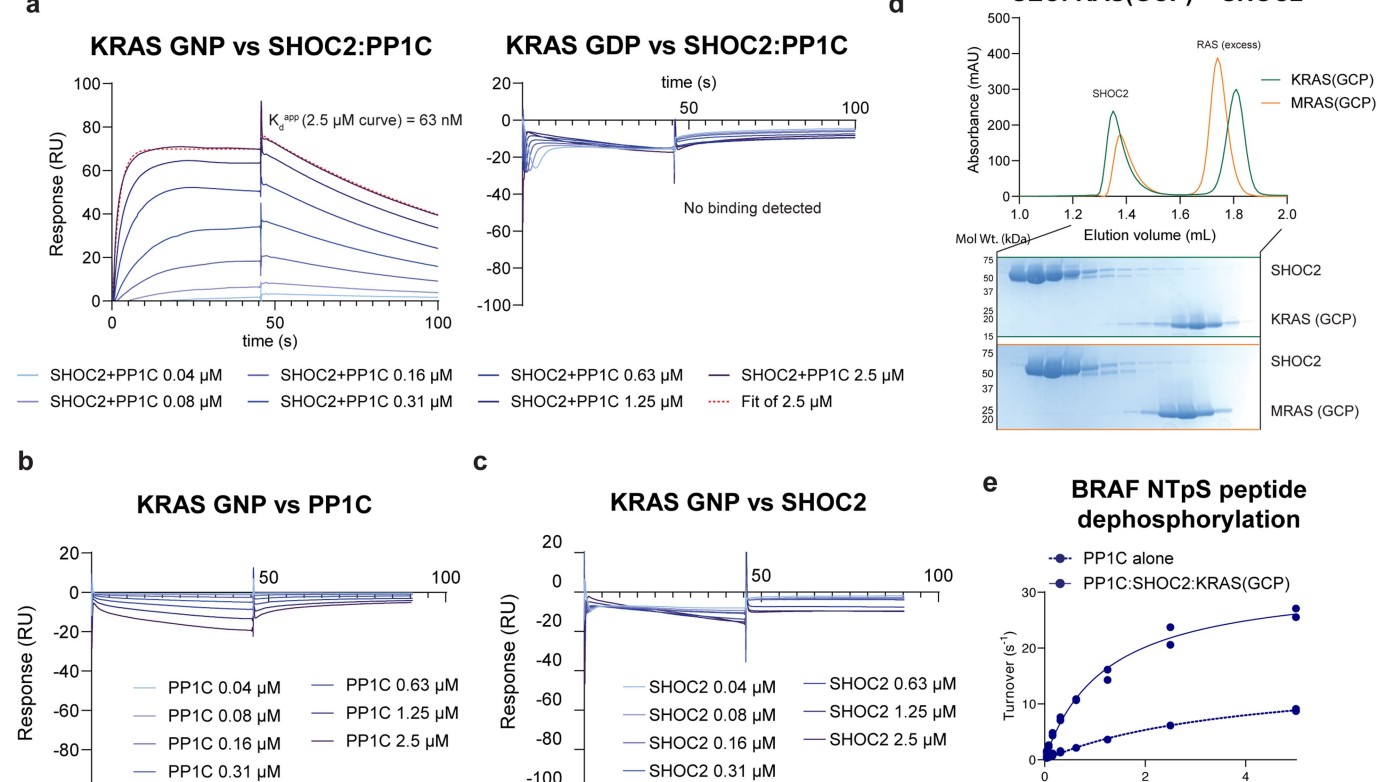

**Extended Data Fig. 2 | SHOC2:PP1C:RAS complex functional validation.**
**a**, SPR data showing that KRAS(GNP) associates with SHOC2:PP1C, but not KRAS(GDP). **b**, Nucleotide bound KRAS does not show binding to PP1C alone nor **c**, SHOC2 alone, showing that binding in **a** is due to RAS binding the SHOC2:PP1C complex. **d**, SEC traces (top) showing SHOC2 does not associate with KRAS(GCP) (green) or MRAS(GCP) (orange) in the absence of PP1C, and associated SDS-PAGE analysis of SEC fractions (bottom). Results representative two independent experiments. **e**, Representative plot of dephosphorylation of BRAF NTpS phosphopeptide by 10 nM PP1C or 10 nM PP1C + 1 μM SHOC2 + 10 μM KRAS(GCP). Data points show 2 and 4 min time points (n = 2).

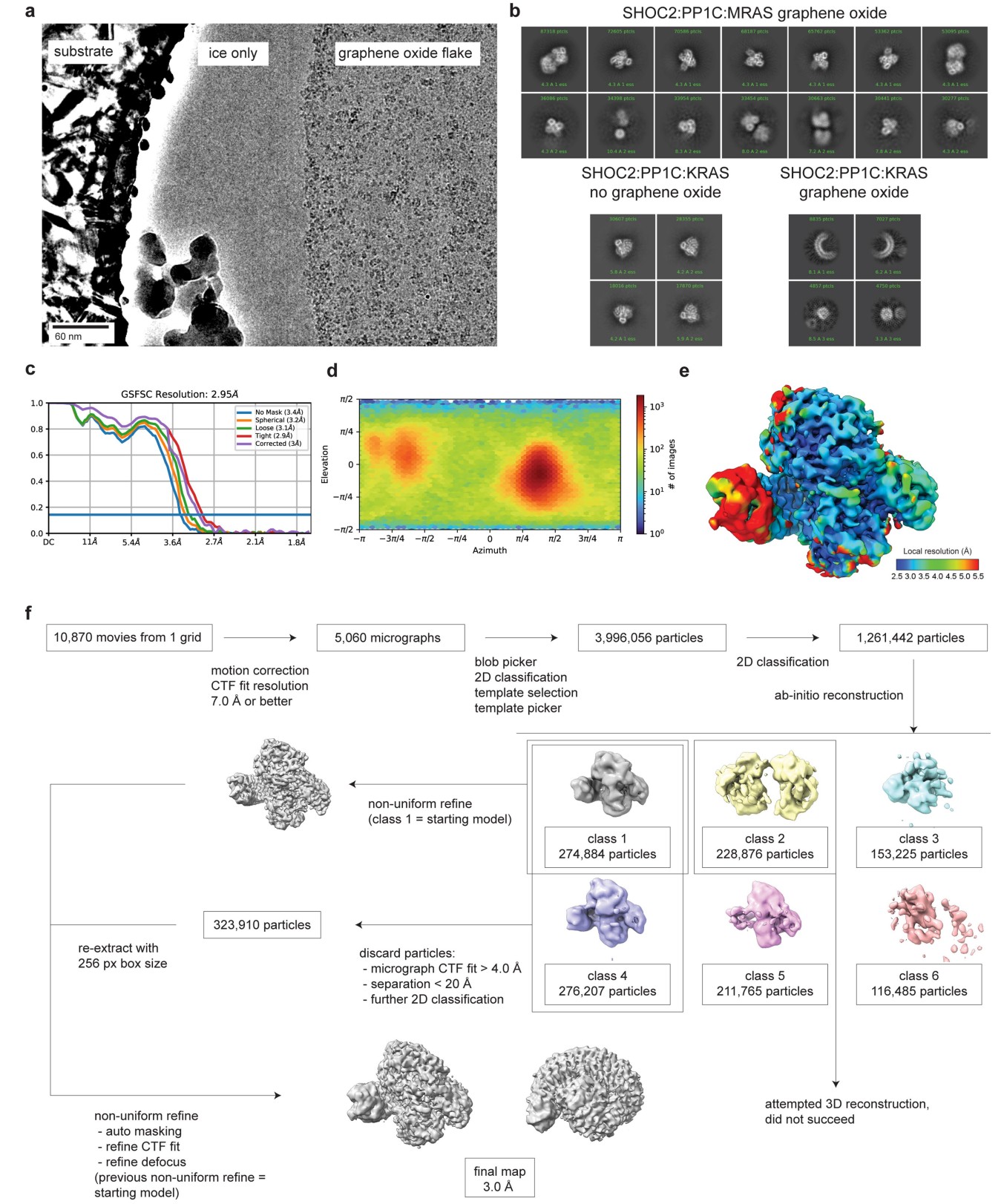

**Extended Data Fig. 3 | Cryo-EM structure of SHOC2:PP1C:MRAS(GCP).**
**a**, Representative micrograph (from 10,870 collected) of SHOC2:PP1C:MRAS (GCP) on graphene oxide containing grids. The far left of the image is gold substrate, the region in the center is vitreous ice, and the right is a single layer of graphene oxide with adsorbed protein complex. **b**, Representative 2D class averages of SHOC2:PP1C:MRAS (top) show well resolved trimeric complex and more poorly resolved hexameric complex, with a range of views on graphene oxide. SHOC2:PP1C:KRAS complex (bottom left) shows only trimeric complex, but preferred orientation without graphene oxide. SHOC2:PP1C:KRAS on graphene oxide (bottom right) shows complex dissociation. **c**, FSC between two half datasets yielded a resolution estimate of 2.95 Å. **d**, Angular distribution map of SHOC2:PP1C:MRAS reconstruction. Particles show a preferred orientation, but sufficient other views were obtained with the use of graphene oxide to obtain a 3D reconstruction. **e**, Local resolution estimate map indicates flexibility of the C terminal portion of the SHOC2 LRR. **f**, cryo-EM data processing scheme, details described in methods.

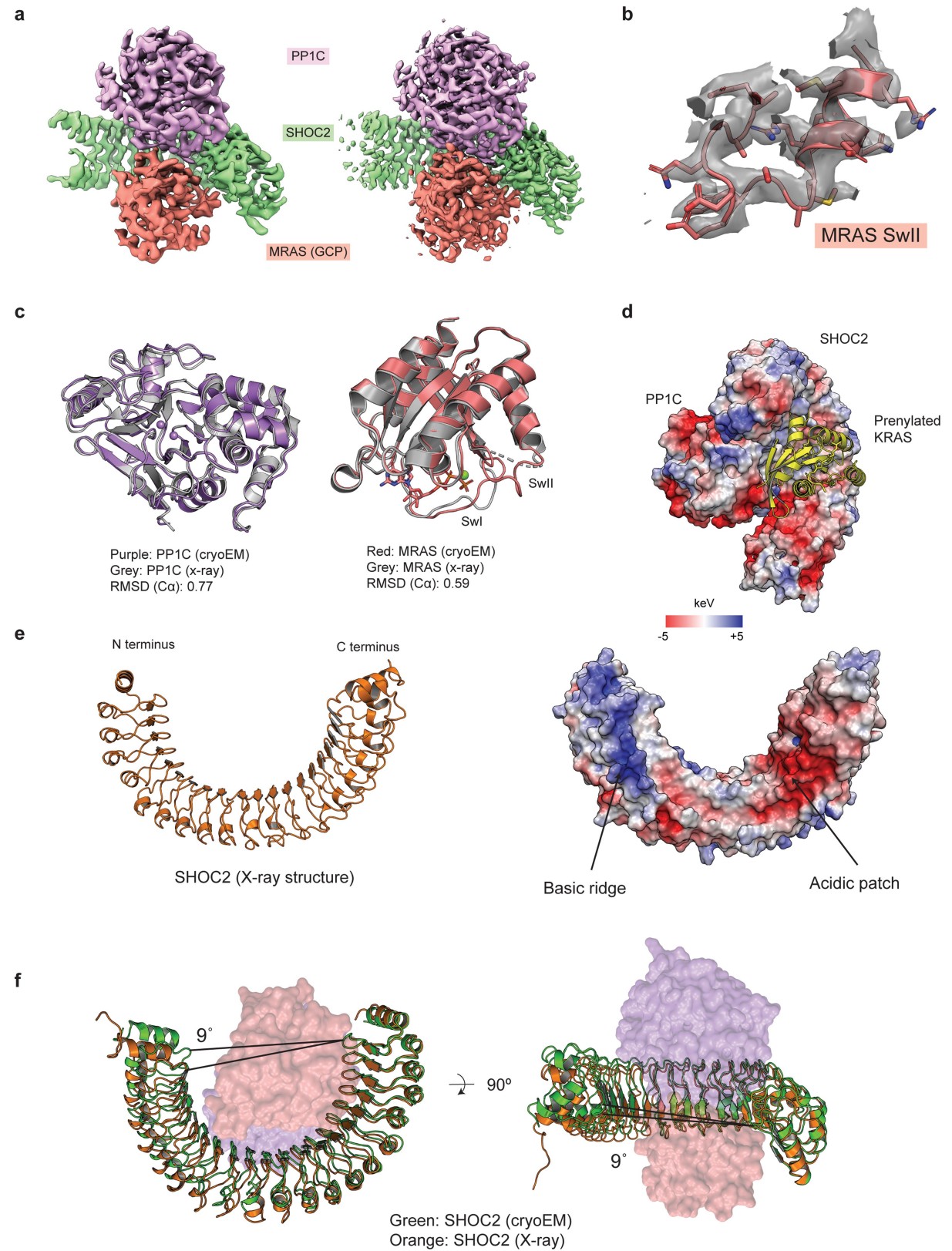

**a**, PP1C

SHOC2

MRAS (GCP)

**b**, MRAS SwII

**c**,
Purple: PP1C (cryoEM)
Grey: PP1C (x-ray)
RMSD (Cα): 0.77

Red: MRAS (cryoEM)
Grey: MRAS (x-ray)
RMSD (Cα): 0.59

SwII

SwI

**d**, SHOC2

PP1C

Prenylated KRAS

keV
−5    +5

**e**, N terminus    C terminus

SHOC2 (X-ray structure)

Basic ridge    Acidic patch

**f**, 9°    90°    9°

Green: SHOC2 (cryoEM)
Orange: SHOC2 (X-ray)

**Extended Data Fig 4 | Cryo-EM and x-ray structure comparisons.**
 **a**, Unsharpened cryoEM map of SHOC2:PP1C:MRAS complex contoured to 0.15 (left). Sharpened cryoEM map of SHOC2:PP1C:MRAS complex contoured to 0.35 (sharpening B factor = −110 Å²)(right). Density is not observable for the SHOC2 C-terminus at these contours. **b**, Example model and sharpened density of MRAS SwII region. **c**, left, PP1C from cryoEM structure (purple) aligned with PP1C x-ray structure (PDB: 4MOV). Right, MRAS(GCP) from cryoEM structure (salmon) aligned with MRAS(GNP) x-ray structure (PDB:1X1S). **d**, Electrostatic view of SHOC2 and PP1C showing basic and acidic regions (blue/white/red) with aligned model of prenylated MRAS model (based on PDB code 5TAR) (yellow), looking down from the membrane, which is parallel to the surface of the page. **e**, X-ray crystal structure of SHOC2 (orange, left) with electrostatic view showing basic and acidic regions (blue/white/red, right). **f**, Alignment of SHOC2:PP1C:MRAS cryoEM structure (green:purple:salmon) with SHOC2 x-ray structure (orange), performed using the N-terminal cap and first four LRRs of SHOC2. A 9° twist is evident between both structures relative to the 1st LRR.

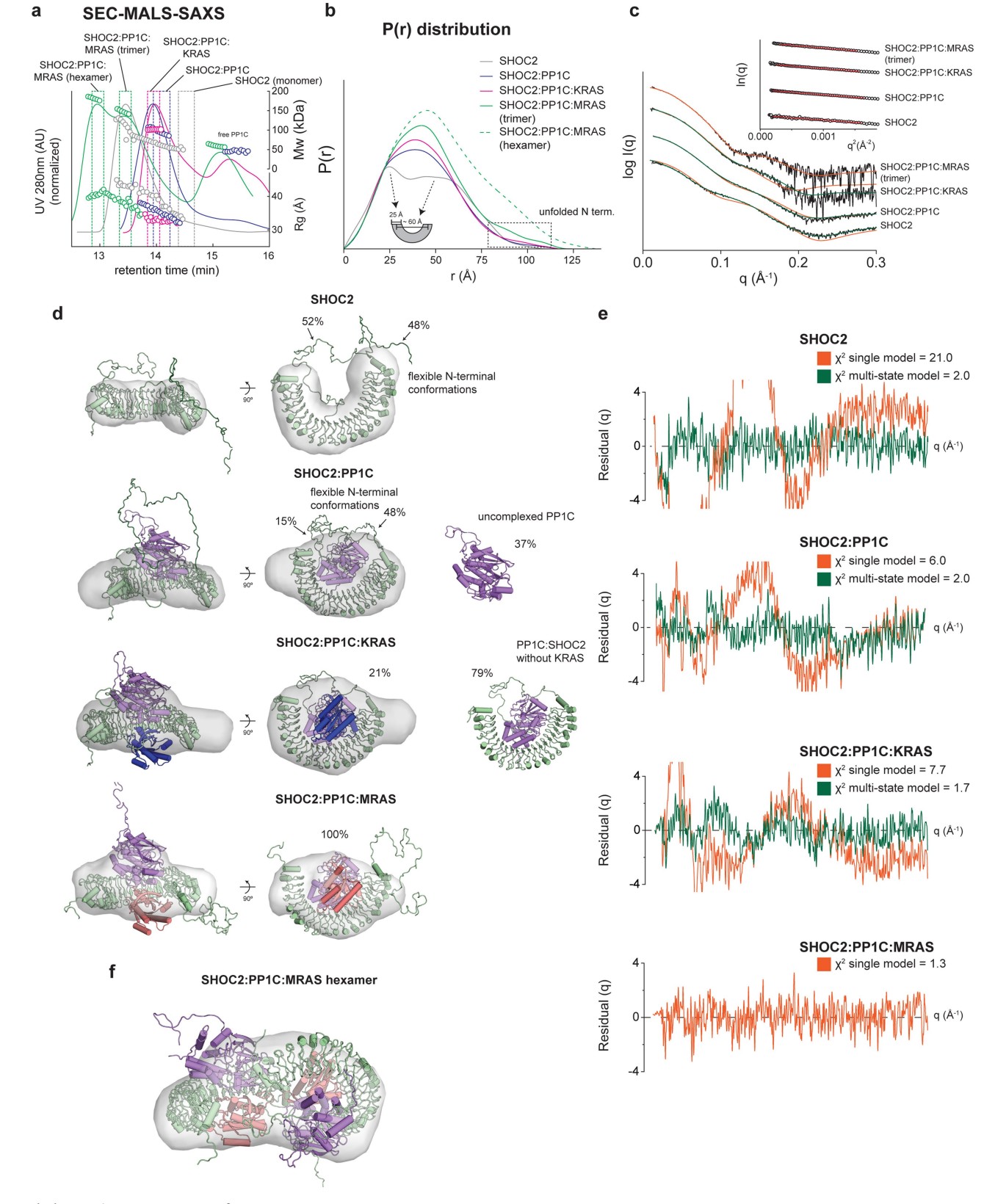

**a** SEC-MALS-SAXS

**b** P(r) distribution

**c**

**d**

**e**

**f**

**Extended Data Fig. 5** | See next page for caption.

**Extended Data Fig. 5 | SAXS analysis and contribution of the SHOC2 N terminus to complex formation. a**, SEC-MALS traces of SHOC2 (grey), SHOC2:PP1C (blue), SHOC2:PP1C:KRAS(GCP) (salmon) and SHOC2:PP1C:MRAS(GCP) (green). SHOC2 alone elutes as a dimer. SHOC2:PP1C:MRAS(GCP) exhibits trimer and hexamer peaks. Dashed rectangles represent frames used for subsequent in-line SAXS analysis. **b**, Distance distribution function (P(r)) for SHOC2 and complexes. SHOC2 exhibits a bimodal distribution showing a horseshoe shape. All proteins show an extended tail at high r values indicative of the unfolded SHOC2 N terminus. **c**, Experimental (black) SAXS curves of SHOC2:PP1C:MRAS, SHOC2:PP1C:KRAS, SHOC2:PP1C, and SHOC2 (top to bottom), overlaid with theoretical SAXS curves for single (orange) and multi-state models (green). Guinier plot for each experimental data (inset). **d**, Molecular models of SHOC2 and complexes (colored ribbon diagrams) overlaid with ab initio SAXS envelopes (grey). The bulge to the side of each molecular envelope represents the extended N-terminus of SHOC2 affecting calculation of the envelope. SHOC2:PP1C, SHOC2:PP1C:KRAS and SHOC2:PP1C:MRAS exhibit similar overall architectures. SHOC2, SHOC2:PP1C and SHOC2:PP1C:KRAS SAXS data are fit by a multi-state model determined by MultiFoXS, indicated by percentage labels, while SHOC2:PP1C:MRAS is well explained by a single state due to stable complex formation. Some dissociation of SHOC2:PP1C was observed in the absence of RAS. **e**, Residuals (($I(q)_{exp} - I(q)_{model}$)/$\sigma(q)$)) of SAXS models compared to data. Multi-state models result in lower $X^2$ values than single state models for SHOC2, SHOC2:PP1C and SHOC2:PP1C:KRAS, whereas a single state model fits the data well for SHOC2:PP1C:MRAS. **f**, SAXS envelope (gray) calculated for SAXS curves of the SHOC2:PP1C:MRAS hexamer. Two atomistic models SHOC2:PP1C:MRAS cryo-EM structure were superimposed manually with the SAXS envelope, showing space exists for a 2:2:2 complex. The exact relative conformations of each molecule cannot be assigned from this data.

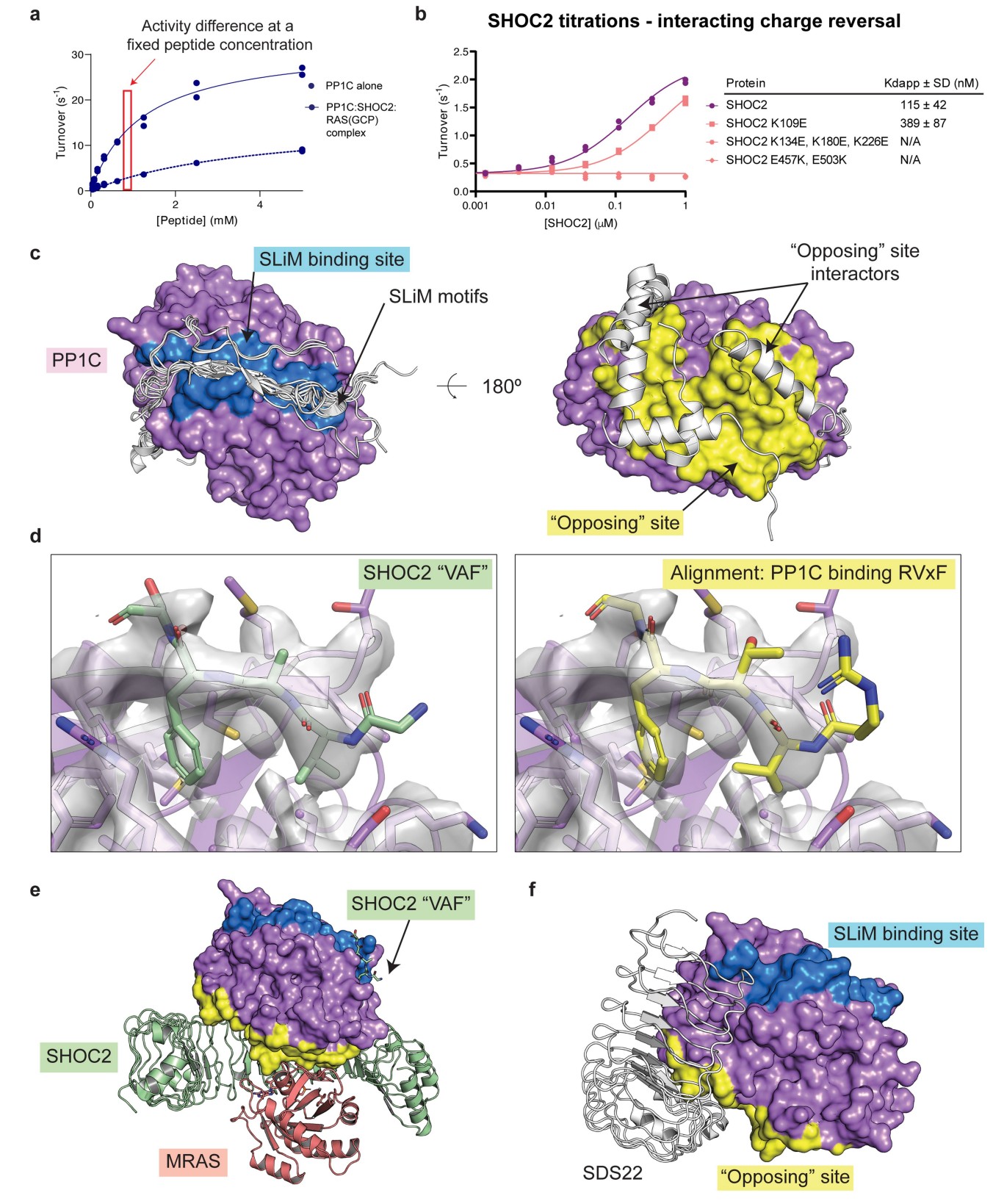

**a**

Activity difference at a fixed peptide concentration

PP1C alone

PP1C:SHOC2:RAS(GCP) complex

**b**

**SHOC2 titrations - interacting charge reversal**

| Protein | Kdapp ± SD (nM) |
|---|---|
| SHOC2 | 115 ± 42 |
| SHOC2 K109E | 389 ± 87 |
| SHOC2 K134E, K180E, K226E | N/A |
| SHOC2 E457K, E503K | N/A |

**c**

SLiM binding site

SLiM motifs

PP1C

"Opposing" site interactors

180°

"Opposing" site

**d**

SHOC2 "VAF"

Alignment: PP1C binding RVxF

**e**

SHOC2 "VAF"

SHOC2

MRAS

**f**

SLiM binding site

SDS22

"Opposing" site

**Extended Data Fig. 6** | See next page for caption.

**Extended Data Fig. 6 | PP1C interaction sites. a**, Overview of technical principle of peptide dephosphorylation based affinity assays. At a fixed concentration of peptide, binding of SHOC2 and RAS will induce a higher PP1C catalytic activity. Data points are from two time points (n = 2) **b**, Representative plot of complex formation with different SHOC2 mutants. [PP1C] = 25 nM, [MRAS(GCP)] = 1 μM. Data points are from two time points (n = 2) (left). Table summarizing $K_D^{app}$ values for different SHOC2 mutants from three independent experiments (n = 3) (right). Mutation of SHOC2 charged patches reduces affinity beyond the limit of detection of the assay. **c**, Left, SLiM binding site (blue) on PP1C (purple), with structure of several interacting SLiMs (white), "Opposing" site (yellow) with structures of several interacting proteins (white).

**d**, Detailed view of the SHOC2 N terminal "VAF" motif (green, left) binding to PP1C with unsharpened cryoEM map (grey surface). Alignment of an existing high resolution x-ray crystal structure containing an RVxF motif bound in this region (yellow, right, from PDB: 5INB) overlays well with the cryoEM map (grey surface). Alignments performed on PP1C. **e**, SHOC2:PP1C:MRAS cryoEM structure with SLiM (blue) and "opposing" (yellow) binding sites highlighted. The majority of SHOC2 interactions with PP1C are away from the SLiM binding and opposing sites. The SHOC2 "VAF" motif binds to the SLiM binding site. MRAS binds solely to the "opposing" site. **f**, SDS22 (white) is another LRR protein which interacts with PP1C, though utilizing different binding surfaces to SHOC2, bridging the SLiM and opposing sites.

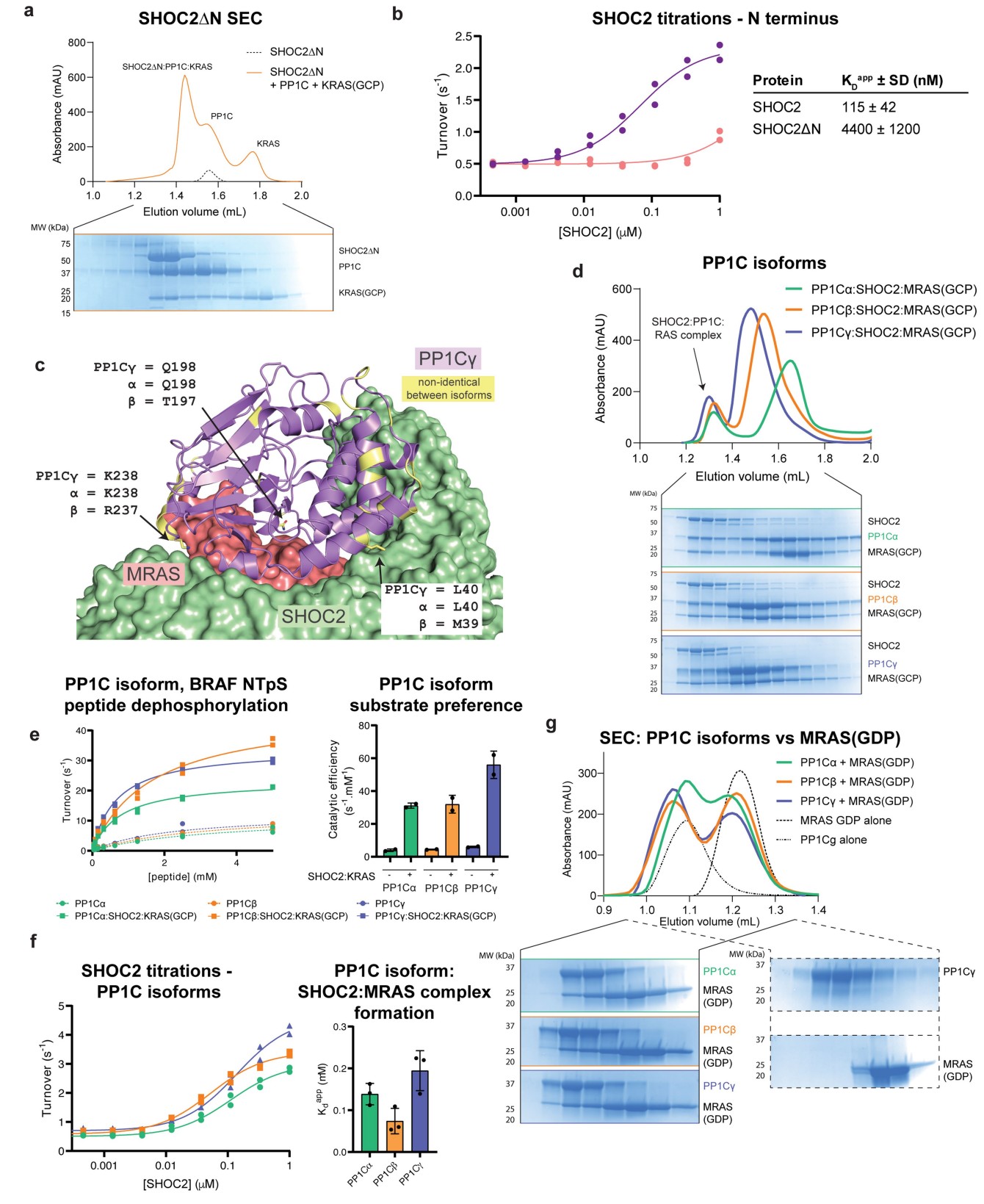

**Extended Data Fig. 7** | See next page for caption.

**Extended Data Fig. 7 | SHOC2:PP1C:RAS interaction data. a**, SEC trace of complex formation between SHOC2ΔN, PP1C and KRAS(GCP) (orange) (top) with associated SDS-PAGE analysis of fractions (bottom). **b**, Peptide dephosphorylation showing $K_D^{app}$ for SHOC2 and SHOC2ΔN. [PP1C] = 25 nM, [MRAS(GCP)] = 1 μM. Data points are from two time points (n = 2) (left). Table summarizing $K_D^{app}$ values for three independent experiments (n = 3) (right). SHOC2ΔN has impaired affinity compared to SHOC2 WT. **c**, Structure of SHOC2:PP1C:MRAS with non-identical residues between PP1C α,β and γ highlighted in yellow. Three non-identical residues located at complex interfaces have similar chemical properties (arrows). **d**, SEC traces and associated SDS-PAGE analysis showing complex formation between SHOC2, MRAS and PP1C α,β and γ isoforms. **e**, Left: Representative peptide dephosphorylation assay against BRAF NTpS with PP1C α,β and γ isoforms. Data points are from two time points (n = 2) (left). Right: Bar graph summarizing mean catalytic efficiency +/- SD from three independent experiments (n = 3). All isoforms show an increase in catalytic efficiency when bound to SHOC2:RAS. [PP1C] = 10 nM, [KRAS(GCP)] = 10 μM. **f**, Left: Representative peptide dephosphorylation assay showing SHOC2 $K_D^{app}$ for MRAS:PP1C α, β and γ complexes. Data points are from two time points (n = 2) (left). Right: Bar graph summarizing mean SHOC2 $K_D^{app}$ +/- SD from three independent experiments (n = 3). [PP1C] = 25 nM, [MRAS(GCP)] = 1 μM. **g**, SEC traces (top) and associated SDS-PAGE analysis (bottom) showing MRAS(GDP) can interact with PP1C α,β and γ isoforms. All SEC and SDS-PAGE data is representative of two independent experiments.

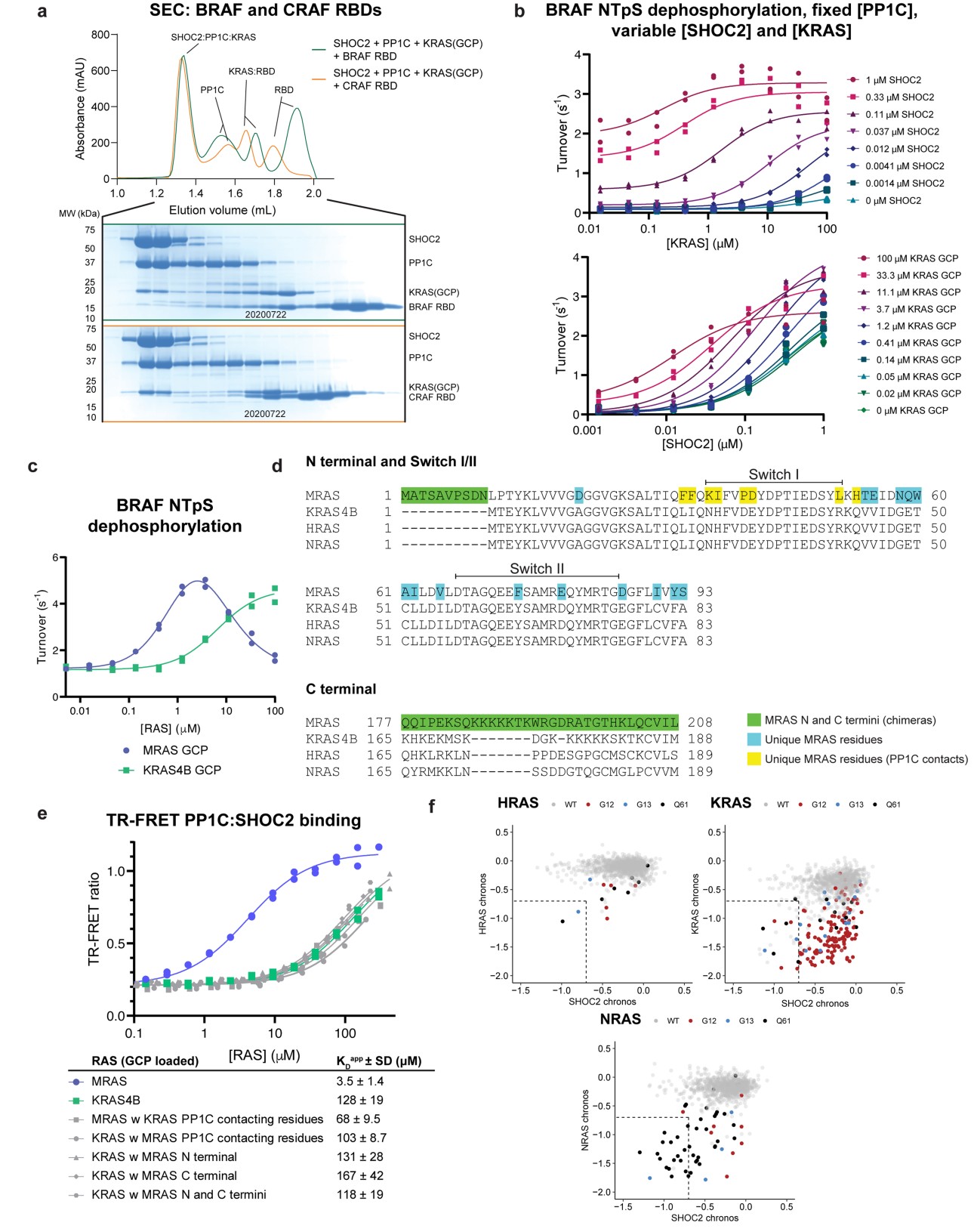

**Extended Data Fig. 8** | See next page for caption.

**Extended Data Fig. 8 | RAS binding interactions. a**, SEC trace showing SHOC2:PP1C:KRAS cannot interact with BRAF RBD (green) or CRAF RBD (orange) (top) with SDS-PAGE analysis of SEC fractions (bottom). A FLAG tag on the CRAF RBD increases its mass slightly compared to the BRAF RBD. Results representative of two independent experiments **b**, BRAF pS365 peptide dephosphorylation by PP1C at varying SHOC2 and KRAS concentrations shows a dependence of the $K_d^{app}$ of one binding partner on concentration of the other. Data points are from two time points (n = 2). **c**, Dephosphorylation of BRAF pS365 peptide (0.2 mM) by PP1C:SHOC2 in the presence of varying RAS concentrations. Data points are from two time points (n = 2). MRAS exhibits a lower $K_D^{app}$ than KRAS, though also shows decreasing activity at higher concentrations, possibly because higher concentrations of the SHOC2:PP1C:MRAS trimer promote formation of a SHOC2:PP1C:MRAS hexamer which may be catalytically inactive. [PP1C] = 10 nM, [SHOC2] = 100 nM. **d**, Sequence alignment between M, K, H and NRAS N termini, Switch I and II (top) and C termini (bottom) **e**, Top: TR-FRET measuring association between PP1C and SHOC2 in the presence of different GCP bound MRAS/KRAS chimeras, showing that 8 residues on MRAS which contact PP1C (see yellow highlighted residues in (**b**)) are critical for high binding affinity. Neither the MRAS N nor C termini improve KRAS binding affinity. The KRAS:MRAS chimera showed a modest but statistically significant binding affinity improvement compared to KRAS WT (103 µM vs 128 µM, p = 0.02), but did not fully recapitulate the strong affinity of MRAS WT, showing that other factors such as MRAS Switch I and II dynamics are also important in driving complex formation. Data points are from measurements taken at two time points (n = 2) (top). Bottom: Table summarizing $K_D^{app}$ values from three independent experiments (n = 3). [Tb-PP1C] = 2 nM, [Red-SHOC2] = 200 nM. **f**, chronos score correlations plotted separately for HRAS, KRAS and NRAS vs SHOC2. Main Fig. 3c represents the combination of these plots, with the lowest of the three H/K/NRAS chronos scores for each cell line selected for display in Fig 3c.

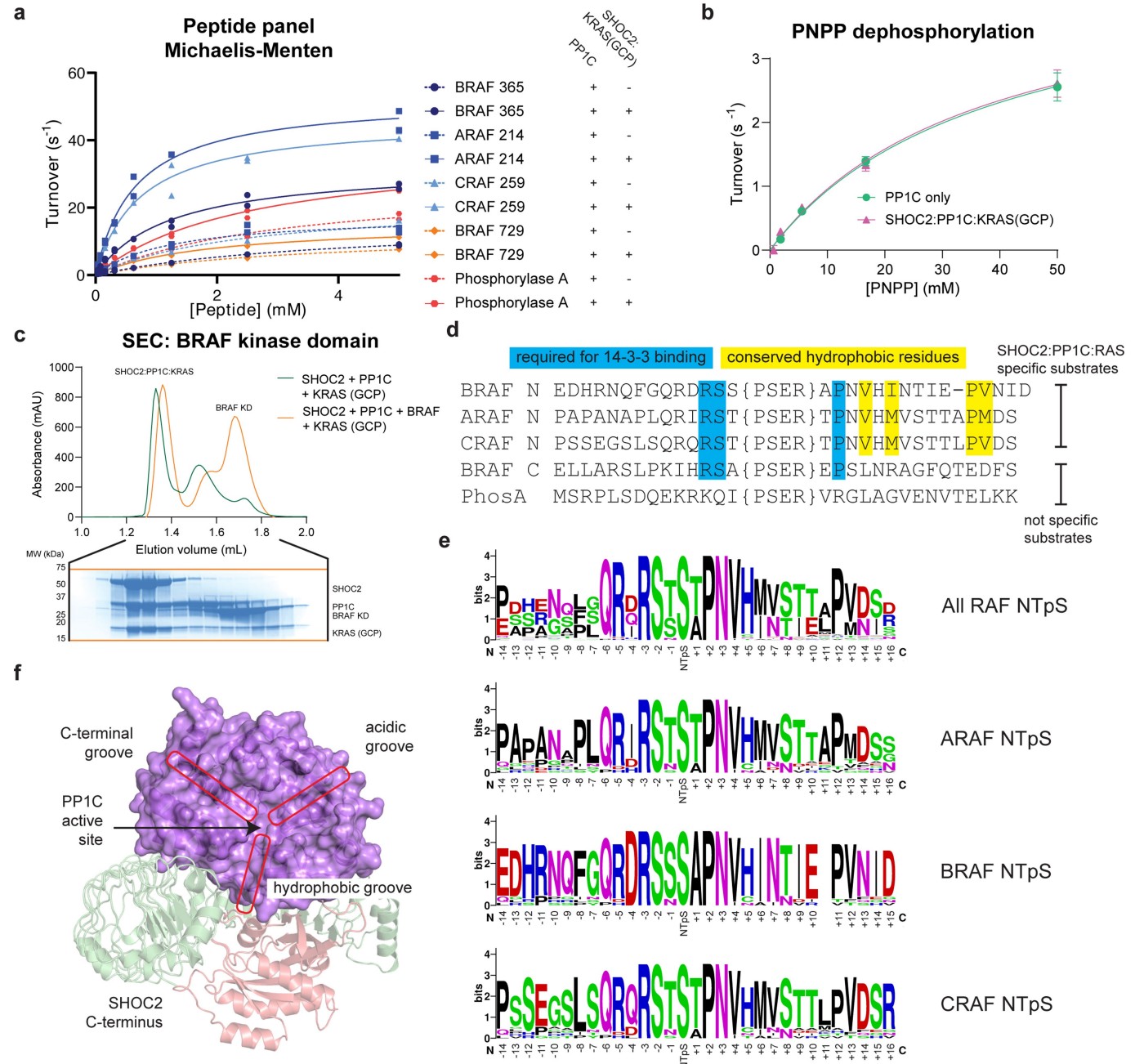

**Extended Data Fig. 9 | PP1C substrate recognition is driven by SHOC2 and RAS. a**, Representative peptide dephosphorylation assay showing higher catalytic efficiency for PP1C when complexed with SHOC2 and KRAS against A/B/CRAF NTpS, but not against Non-Target Peptides. Data points are from two time points (n = 2). See Fig. 4 for statistical summary of three independent experiments. [PP1C] = 10 nM, [KRAS(GCP)] = 10 μM. **b**, Peptide dephosphorylation assay showing PP1C (green) or PP1C:SHOC2:KRAS (maroon) dephosphorylation of p-Nitrophenyl Phosphate (PNPP). Error bars represent mean +/- SD of four time points (n = 4). [PP1C] = 10 nM, [KRAS(GCP)] = 10 μM. Results are representative of two independent experiments. **c**, SEC analysis showing no association between SHOC2:PP1C:KRAS and BRAF kinase domain (orange) (top) with associated

SDS-PAGE analysis (bottom). Results representative of two independent experiments **d**, Sequence alignment of A/B/CRAF NTpS (expected biological targets of SHOC2:PP1C:RAS), CRAF CTpS and Phosphorylase A pS15 (Non-Target Peptides). NTpS peptides contain C terminal hydrophobic residues (yellow) and both NTpS and CTpS contain critical 14-3-3 interacting residues (blue) which may have evolved for 14-3-3 recognition, rather than SHOC2:PP1C:RAS recognition. **e**, Sequence logo showing conservation among A/B/CRAF NTpS across 18 representative species from *C. elegans* to *H. sapiens* (where equivalent RAF isoforms were available) (top), or separate RAF NTpS across species (bottom three). **f**, SHOC2:PP1C:MRAS structure with PP1C substrate binding grooves highlighted (red ovals).

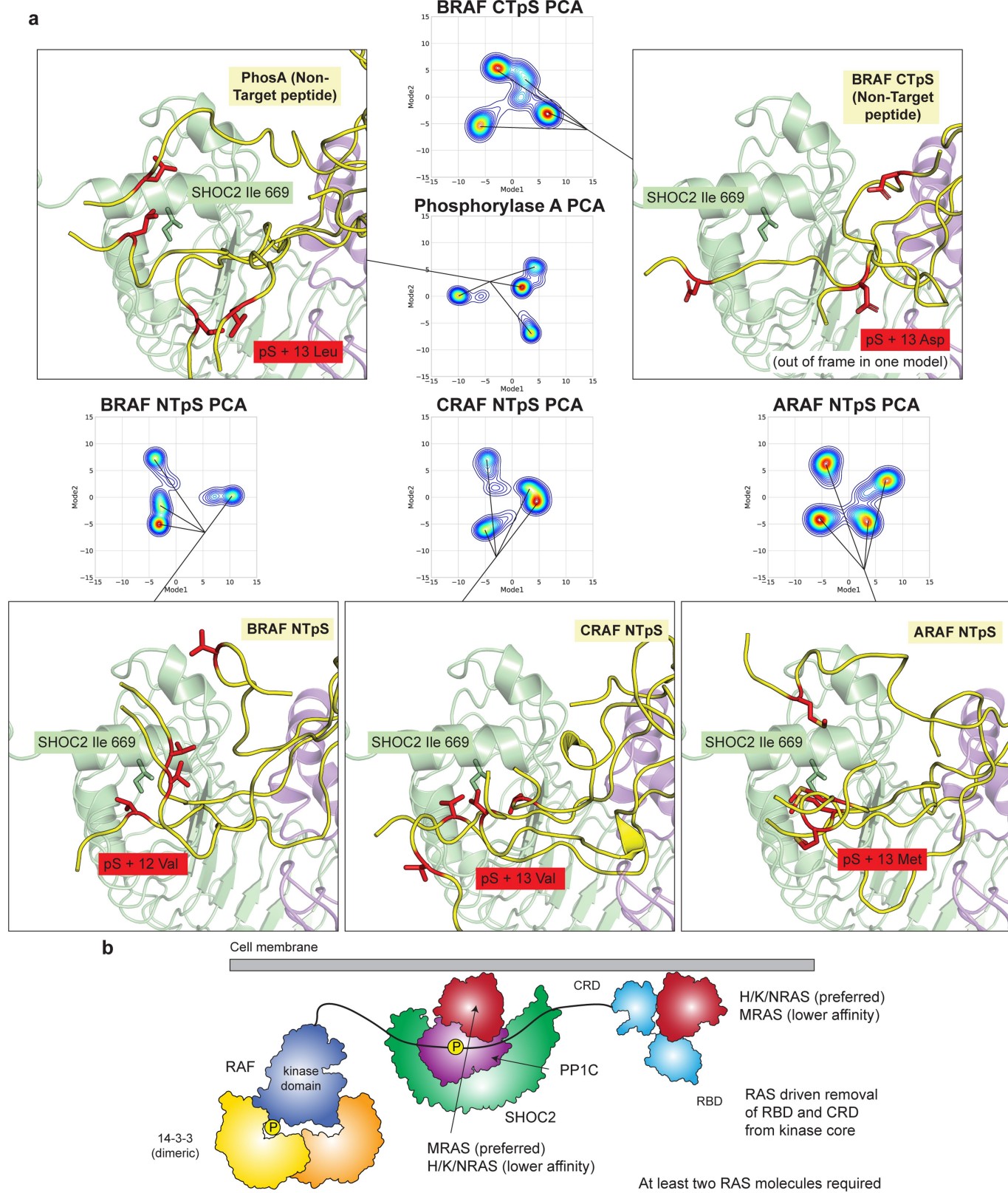

**Extended Data Fig. 10 | Molecular dynamics simulations of SHOC2:PP1C:RAS interactions with peptides. a**, Principal component analysis (PCA) of each peptide:protein simulation showing clusters of stable states within each simulation. Representative structures were chosen from within the most populated states, as indicated (black lines). Representative structures were aligned based on the final LRR and C-terminal capping helices of SHOC2 to place the SHOC2 hydrophobic groove on a common view. Only a single SHOC2:PP1C:MRAS complex is shown for clarity. RAF NTpS peptides interact with the SHOC2 hydrophobic groove in a majority of simulations, but not for the Non-Target Peptides. Hydrophobic interactions likely consist of a considerable entropic portion, preventing accurate calculations for free energy of binding for these models. **b**, Model of RAS binding. Under normal conditions, H/K/NRAS is the preferred binding partner of RAF, whilst MRAS is preferred for SHOC2:PP1C. RAF and SHOC2:PP1C:RAS co-localize at the membrane prior to substrate dephosphorylation.

**Extended Data Table 1 | CryoEM data collection, refinement and validation statistics for the SHOC2:PP1C:MRAS complex**

|  | SHOC2:PP1C:MRAS (EMDB-25044) (PDB 7DS0) |
|---|---|
| **Data collection and processing** | |
| Magnification | 105,000x |
| Voltage (kV) | 300 |
| Electron exposure (e–/Å$^2$) | 64 |
| Defocus range (μm) | 0.5-1.5 |
| Pixel size (Å) | 0.419 |
| Symmetry imposed | C1 |
| Initial particle images (no.) | 3,996,056 |
| Final particle images (no.) | 323,910 |
| Map resolution (Å) | 2.95 |
| FSC Threshold | 0.143 |
| Map resolution range (Å) | 32.2.95 |
| | |
| **Refinement** | |
| Initial model used (PDB code) | 7DS1 (SHOC2), 1X1S (MRAS), 4MOV (PP1C) |
| Model resolution (Å) | 2.95 |
| FSC Threshold | 0.5 |
| Model resolution range (Å) | 32-2.95 |
| Map sharpening *B* factor (Å$^2$) | -109 |
| Model composition | |
| Non-hydrogen atoms | 7730 |
| Protein residues | 965 |
| Ligands | 4 |
| *B* factors (Å$^2$) | |
| Protein | 140.2 |
| Ligand | 123.7 |
| R.m.s. deviations | |
| Bond lengths (Å) | 0.008 |
| Bond angles (°) | 0.939 |
| Validation | |
| MolProbity score | 1.94 |
| Clashscore | 11.44 |
| Poor rotamers (%) | 0.91 |
| Ramachandran plot | |
| Favored (%) | 94.7 |
| Allowed (%) | 5.3 |
| Disallowed (%) | 0 |

**Extended Data Table 2 | Data collection and refinement statistics for the SHOC2 crystal structure**

|  | SHOC2 (PDB: 7DS1) |
| --- | --- |
| **Data collection** | |
| Space group | I 2 2 2 |
| Cell dimensions | |
| $a, b, c$ (Å) | 168.63, 201.83, 233.77 |
| $\alpha, \beta, \gamma$ (°) | 90, 90, 90 |
| Resolution (Å) | 48.06-3.19 (3.39-3.19)* |
| $R_{sym}$ | 0.17 (3.38) |
| $I / \sigma I$ | 8.94 (0.51) |
| CC ½ (%) | 99.9 (29.4) |
| Completeness (%) | 99.6 (98.8) |
| Redundancy | 6.7 (7.0) |
|  | |
| **Refinement** | |
| Resolution (Å) | 3.19 |
| No. reflections | 444449 |
| $R_{work}/R_{free}$ | 0.210/0.238 |
| No. atoms | 15721 |
| Protein | 15721 |
| Ligand/ion | 0 |
| Water | 0 |
| $B$-factors | 120.81 |
| Protein | 120.81 |
| Ligand/ion | N/A |
| Water | N/A |
| R.m.s. deviations | |
| Bond lengths (Å) | 0.005 |
| Bond angles (°) | 0.83 |

# Reporting Summary

## Statistics

For all statistical analyses, confirm that the following items are present in the figure legend, table legend, main text, or Methods section.

| n/a | Confirmed | |
|---|---|---|
| ☐ | ☒ | The exact sample size (*n*) for each experimental group/condition, given as a discrete number and unit of measurement |
| ☐ | ☒ | A statement on whether measurements were taken from distinct samples or whether the same sample was measured repeatedly |
| ☐ | ☒ | The statistical test(s) used AND whether they are one- or two-sided *Only common tests should be described solely by name; describe more complex techniques in the Methods section.* |
| ☒ | ☐ | A description of all covariates tested |
| ☒ | ☐ | A description of any assumptions or corrections, such as tests of normality and adjustment for multiple comparisons |
| ☐ | ☒ | A full description of the statistical parameters including central tendency (e.g. means) or other basic estimates (e.g. regression coefficient) AND variation (e.g. standard deviation) or associated estimates of uncertainty (e.g. confidence intervals) |
| ☐ | ☒ | For null hypothesis testing, the test statistic (e.g. *F*, *t*, *r*) with confidence intervals, effect sizes, degrees of freedom and *P* value noted *Give P values as exact values whenever suitable.* |
| ☒ | ☐ | For Bayesian analysis, information on the choice of priors and Markov chain Monte Carlo settings |
| ☒ | ☐ | For hierarchical and complex designs, identification of the appropriate level for tests and full reporting of outcomes |
| ☐ | ☒ | Estimates of effect sizes (e.g. Cohen's *d*, Pearson's *r*), indicating how they were calculated |

*Our web collection on statistics for biologists contains articles on many of the points above.*

## Software and code

Policy information about availability of computer code

| Data collection | SerialEM 3.7.11. |
|---|---|
| Data analysis | Prism8 (GraphPad); UCSF ChimeraX 1.2.5; Phenix 1.18; The ConSurf Server (https://consurf.tau.ac.il/); PyMOL 2.3.2 (The PyMOL Molecular Graphics System, Schrödinger, LLC); Coot 0.89; UniDec 4.4.0; CryoSPARC 3.0.1; CryoSPARC Live 3.1; MODELLER 9.21; BILBOMD 1.0; FoXS 2.1; FF19SB; Gaussian09; Amber2018; CPPTRAJ 18.0. |

For manuscripts utilizing custom algorithms or software that are central to the research but not yet described in published literature, software must be made available to editors and reviewers. We strongly encourage code deposition in a community repository (e.g. GitHub). See the Nature Portfolio guidelines for submitting code & software for further information.

## Data

Policy information about availability of data

All manuscripts must include a data availability statement. This statement should provide the following information, where applicable:

- Accession codes, unique identifiers, or web links for publicly available datasets
- A description of any restrictions on data availability
- For clinical datasets or third party data, please ensure that the statement adheres to our policy

Coordinates and related data for SHOC2 and SHOC2:PP1C:MRAS complex have been deposited in the PDB and EMDB respectively: PDB 7DS1 for SHOC2 and PDB 7DS0 and EMDB-25044 for the SHOC2:PP1C:MRAS complex. SAXS data and atomistic models have been deposited at the SASBDB database as entries SDSDMB5 (SHOC2), SDSDMC5 (SHOC2:PP1C), SDSDMD5 (SHOC2:PP1C:KRAS) and SASDME5 (SHOC2:PP1C:MRAS).

DepMap release Public 21Q4 datasets containing cell line information (sample_info.csv), chronos scores (CRISPR_gene_effect.csv) and mutational status

# Field-specific reporting

Please select the one below that is the best fit for your research. If you are not sure, read the appropriate sections before making your selection.

☒ Life sciences    ☐ Behavioural & social sciences    ☐ Ecological, evolutionary & environmental sciences

For a reference copy of the document with all sections, see nature.com/documents/nr-reporting-summary-flat.pdf

# Life sciences study design

All studies must disclose on these points even when the disclosure is negative.

| | |
|---|---|
| Sample size | For enzyme assays, n=2 was chosen for technical replicates to ensure linearity of reactions over time. For binding assays, n=2 was chosen for technical replicates to ensure binding equilibrium had been reached. For X-ray structure determination, only one diffracting crystal was ever obtained for analysis. For CryoEM data collection, sample size was not pre-determined. The number of micrographs collected was determined by microscope availability and sufficient resolution being reached in the final structure to make functional conclusions. |
| Data exclusions | All attempts at replication were successful. No replicates were excluded from analysis for any experiments. |
| Replication | Biochemical assays were performed as three independent experiments. Within each independent experiment two time points were samples (technical replicates). |
| Randomization | For the X-ray structure, a random Rfree set of data was excluded for later validation of the model. For the CryoEM structure, two random half data sets were independently refined to provide resolution cutoff estimates. |
| Blinding | Blinding is not applicable to this study. Blinding was not performed in this study because it is not necessary or practical for X-ray or CryoEM structural determination. |

# Reporting for specific materials, systems and methods

We require information from authors about some types of materials, experimental systems and methods used in many studies. Here, indicate whether each material, system or method listed is relevant to your study. If you are not sure if a list item applies to your research, read the appropriate section before selecting a response.

## Materials & experimental systems

| n/a | Involved in the study |
|---|---|
| ☒ | ☐ Antibodies |
| ☐ | ☒ Eukaryotic cell lines |
| ☒ | ☐ Palaeontology and archaeology |
| ☒ | ☐ Animals and other organisms |
| ☒ | ☐ Human research participants |
| ☒ | ☐ Clinical data |
| ☒ | ☐ Dual use research of concern |

## Methods

| n/a | Involved in the study |
|---|---|
| ☒ | ☐ ChIP-seq |
| ☒ | ☐ Flow cytometry |
| ☒ | ☐ MRI-based neuroimaging |

# Eukaryotic cell lines

Policy information about cell lines

| | |
|---|---|
| Cell line source(s) | Sf9 cell line (insect cells) was used for over-expression of some of the proteins used in this study which were then purified from the cell using chromatography techniques. Sf9 cells were obtained from Expression Systems (https://expressionsystems.com/) |
| Authentication | The cell line was not authenticated independently. |
| Mycoplasma contamination | The cell line was not tested for mycoplasma contamination. |
| Commonly misidentified lines (See ICLAC register) | Not Applicable |

