## [Peer Review File · Nature]

Manuscript Title: Structural basis for SHOC2 modulation of RAS signaling

Reviewer Comments & Author Rebuttals

Reviewer Reports on the Initial Version:

Referees' comments:

Referee #1 (Remarks to the Author):

The MRAS/SHOC2/PP1C complex is well-known for its ability to dephosphorylate the key NTPS 14-3-3 binding site on RAF proteins, thereby contributing to RAF activation. A structural understanding of the event is currently lacking and the authors set to solve this issue by elucidating the structure of the complex by cryoEM. Briefly, using recombinant proteins, they first demonstrated the requirement for MRAS GTP-loading for complex formation as assessed by size exclusion chromatography. Intriguingly, unlike reports by other groups, they also found that RAS oncoproteins (H, K and NRAS) also allowed RAS/SHOC2/PP1C complex formation in a GTP-dependent manner, albeit these proteins exhibited significantly lower binding affinities compared to MRAS. Using cryoEM, they then solved the structure of the MRAS/SHOC2/PP1C complex at an overall resolution of 3.0 Å. Preliminary structural information was also gathered on the KRAS/SHOC2/PP1C complex. Although too early to be conclusive, this latter complex seems reminiscent of the MRAS/SHOC2/PP1C complex, which would suggest similar mechanism of action across the various RAS-containing SHOC2/PP1C complexes. They next described the binding interfaces linking MRAS, SHOC2 and PP1C. This provided structural insights into the binding requirements, the impact of GTP binding, substrate recognition, and the underlying basis of RASopathy mutations peppering the MRAS, SHOC2 and PP1C proteins. Finally, they note that the structure of the MRAS/SHOC2/PP1C complex offers novel therapeutic opportunities.

Overall, this is an exciting and well-written article that provides genuinely novel and insightful information on a key regulatory complex of the RAS/MAPK pathway. Although there is already a wealth of data on the topic, the current work provides at last a conclusive structural framework for their interpretation. That said, in my opinion, there are a few issues that need to be addressed to strengthen the data and eliminate some of the more speculative aspects of the work that currently reduce its significance.

Main points:

1- Each binding interface should be validated by site-directed mutagenesis followed by binding and functional assays. Although some is done indirectly by mentioning mutations found elsewhere (i.e. RASopathies or genetic screen), they should do it systematically for each surface. For instance, they should address the functional relevance of the VAF motif.

2- line 167: They claim that the few amino acid differences between PP1C isoforms likely do not explain

their distinct biological or binding affinities to SHOC2. They should address this point directly either by testing (binding and function) relevant amino acid changes or by comparing the binding affinities of the different PP1C isoforms.

3- Given the significantly higher binding affinity of MRAS compared to H,K,NRAS GTPases, compounded with the fact that other groups failed to detect significant binding of non-MRAS GTPases to SHOC2/PP1C, the relevance of H,K,NRAS forming complexes with SHOC2/PP1C in normal physiological conditions remains uncertain and needs to be addressed. A- Do any of the H,K,NRAS/SHOC2/PP1C complexes endogenously exist in cells? B- Are they genuinely involved in RAF NTPS dephosphorylation in the absence of MRAS? Otherwise, their results remain speculative.

4- PP2A has also been reported as a phosphatase involved in dephosphorylating the NTPS sites in RAF proteins following RAS activation (Ory et al. Curr Biol 2003). Is it possible that only MRAS forms physiological complexes with SHOC2/PP1C, whereas H,K,NRAS use other mechanisms, such as PP2A-dependent mechanisms?

5- Since RAS apparently uses overlapping interfaces for binding RAF and SHOC2/PP1C, an interesting possibility is that two distinct RAS molecules are required for driving RAF activation, namely, one that recruits RAF through RBD binding, and one that recruits SHOC2/PP1C to dephosphorylate RAF's NTPS site. This insight is conceptually novel and likely applicable to other small GTPases involved in other complexes/signaling mechanisms. Unfortunately, the concept remains speculative at this point. A formal demonstration would strongly strengthen the reach of their work. As a corollary to this point, could H,K,NRAS work in concert with MRAS for optimal RAF activation? For example, one class of GTPases (H,K,NRAS) might serve to recruit RAF, while the other (MRAS) serves to recruit SHOC2/PP1C?

Referee #2 (Remarks to the Author):

In this manuscript Liau et al., present a cryo-EM structure of the SHOC2-PP1C-RAS complex, as well as additional structural and biochemical data on the regulation of BRAF dimerization and activation by the SHOC2:PP1C:RAS complex. The reported structures, accompanied by the Size Exclusion Chromatography (SEC) and Surface Plasmon Resonance (SPR) data support a model by which a) formation of the complex requires GTP-bound RAS, b) MRAS has higher affinity for binding in the complex, compared to K/N/HRAS and c) even though a peptide containing S365 readily binds and is dephosphorylated by the complex, ordered domains of BRAF are not able to bind it. The data indicate that two different RAS molecules are required for BRAF activation in the membrane, one that binds the RBD domain of BRAF and the other that participates in the SHOC2:PP1:RAS complex. The data are convincing and consistent with a large body of previous literature (Rodriguez-Viciano et al, Mol Cell, 2006, as well as more recent studies on the topic by Young et al., PNAS 2018, Boned del Rio et al PNAS, 2019), to which they provide additional structural and biochemical support. An important question that remains unresolved in the field is to define the contribution of mutationally activated H/N/KRAS in relation to MRAS in regulating BRAF S365

dephosphorylation in cells. To what extent oncogenic K/N/HRAS mutations are sufficient to bypass the need for MRAS or SHOC2 activity? Answering these questions may have important translational implications in terms of identifying potentially new therapeutic approaches to target MAPK signaling. These questions are only indirectly addressed in the manuscript, without cell-based experimental evidence.

Further, the manuscript would be strengthened with experimental validation of some critical predictions: The authors prepared chimeric KRAS with MRAS N and or C termini and did not observe increased binding to the SHOC2:PPC1 compared to KRAS. They could prepare based on the structure, MRAS in which critical amino acids are replaced with the analogous of KRAS, and show weaker binding to the complex. Similarly, in Fig 4, reported loss of function and gain of function mutations in SHOC2, PP1C and MRAS are only explained using the generated structures, but not experimentally validated in the biochemical assays the authors have in hand.

Other comments:

- Fig 1b is the only main fig. panel that presents the cryo-EM density throughout the paper (together with the local res representation in ext fig. 3d). Additional representations would improve the presentation of the data. For example the reconstruction can also be shown in higher threshold so that structural elements can be more readily recognized. Moreover, a figure showing fit of the atomic model to the cryo-EM density, with representative structural elements shown in closeup, is needed.
- An angular distribution plot is not presented for the data and it is needed given that issues of preferred orientation for some of the complexes are discussed.
- In ext Fig. 3a, the “representative” micrograph shown exhibits severe issues with particle distribution. The rationale for presenting this micrograph is unclear as it generally would be considered very poor. Were all micrographs in the dataset of similar appearance or could a “better” micrograph be presented? In addition, an inset in higher magnification is necessary to present a better view of the particles in the micrograph.
- The 2D class averages presented need to be increased in size, as their small size significantly hinders a proper evaluation of their quality.
- Was any masking used with local refinement or 3D classification to try to improve the parts of the protein that seem to be more flexible (lower resolution towards the ends in the local res map)?

Overall, although the main conclusions of the manuscript are not entirely surprising, given the current biochemical and cell-based conceptual understanding on how the SHOC2-PP1C-MRAS complex dephosphorylates and stabilizes dimeric RAF, the study provides the first cryo-EM structure of a complex critical for MAPK activation, as well as useful structural and biochemical information.

Referee #3 (Remarks to the Author):

The RAS-RAF axis is critical for many biological/pathological processes, in particular tumorigenesis. This

work by Liau et al. demonstrates RAS isoform specificity and its GTP dependence for formation of the RAS-SHOC2-PP1 complex, and reports a high-resolution cryo-EM structure of this important complex. Main biochemical and structural results in this manuscript appear novel, technically solid and interesting. However, more biochemical studies and maybe more structural characterization are needed to clarify some important aspect of this work.

Main issues:

1. The most surprising aspect of the current RAS-SHOC2-PP1 complex structure is that RAS SwI and SwII are directly involved in its interaction with SHOC2/PP1, and thus RAF cannot be directly recruited to the RAS-SHOC2-PP1 complex for dephosphorylation by PP1. Interestingly, while no apparent binding could be detected, the RAS-SHOC2-PP1 complex has a significantly ($\sim 18X$) lower K_m for the BRAF pS365 peptide than PP1 per se. How is the specificity and lower K_m for the RAF pS365-peptide achieved at the PP1 active site in the complex? How about other RAF phosphor-peptides (e.g. pS259 RAF1, pS214 ARAF)? I wonder if authors have tried to visualize how the RAF pS365-peptide can be recognized by the phosphatase active site of the RAS-SHOC2-PP1 holoenzyme. One of possible ways to stabilize the transient BRAF pS365-peptide substrate binding for cryo-EM analysis is to fuse the RAF phosphor-peptide covalently to the C-terminus of PP1C using the intein protein fusion approach. If cryo-EM analysis of a RAF phosphor-peptide containing complex is not possible, a molecular dynamics simulation of the RAF phospho-peptide recognition by this holoenzyme complex, based on current cryo-EM structure, is needed to obtain a somewhat clear picture on how the MRAS-SHOC2-PP1 complex promotes RAF dephosphorylation and activation. Are RAS and SHOC2 directly involved in RAF substrate recognition or merely provide a membrane anchor for PP1? This issue is not only critical for mechanistic understanding but also for developing novel RAS/RAF inhibitors targeting at the RAF recognition site.
2. Why was the dimer of MRAS-SHOC2-PP1 trimers not further structurally characterized? Could this (MRAS-SHOC2-PP1)₂ hexamer be biologically important? It is hard to see any 3D structural features of this interesting hexamer in the Extended Fig 5f. Could this low-resolution hexameric model be validated by biochemical/biophysical analysis (e.g. cryo-EM, cross-link MS)? Analysis of cross-validated low-resolution hexameric model can be very insightful.
3. Due to flexibility of the SHOC2 scaffold, authors cannot conclude that clinical KRAS inhibitors could lead to steric clashes and thus inhibit/modulate RAS's interactions with the SHOC2-PP1 complex and subsequent RAF dephosphorylation (Fig 4b). Experimental data must be provided to support the claims. Alternatively, the last section of Results (lines 301-312) can be removed.
4. Previous works have identified a number of gain- and loss-of-function mutants, which are associated with RASopathy phenotypes and can be mapped to the RAS-SHOC2-PP1 interfaces. Can some of the key mutants experimentally tested to validate structural observations?

Minor points:

1. Is the described flexibility of SHOC2 scaffold a result of collective flexibility cross the entire LRR structure or a hinge-like motion? It has been previously predicted that SHOC2 contains a hinge in the middle of its LRR fold (Kwon et al, 2021, MCB). A more detailed structural comparison between the RAS-SHOC2-PP1 cryo-EM structure and the SHOC2 crystal structure may help.
2. There is a lack of description, in fig legends or methods, on how 3D structural alignment or

- superposition is done. Were all shown results from globally (or subunit/region-guided) superposition?
3. Can MRAS(GDP) bind to other PP1C isoforms?
 4. Use of GCP and GNP nomenclature should be unified. GMP-PNP was also used in this ms.
 5. In Fig 1, it would be useful to highlight the MRAS SwI and SwII (e.g. in red) in the overall structure;
 6. Comparison of areas of buried surfaces for each interface?
 7. As the SHOC2 scaffold has some structural flexibility as shown by authors, 'steric clash' (line 180) is not a good explanation for a lack of binding of the GDP-bound Ras.
 8. In many cases, including Fig 1 and 4c, molecular labels can be color-coded – same as the colors of corresponding structures.
 9. Crystallographic statistics for the highest resolution data bin do not look good (Table S1). The resolution should be cut to a lower resolution (3.3 or 3.4?).
 10. How does SCRIB play a regulatory role for the RAS-SHOC2-PP1 complex?

Author Rebuttals to Initial Comments:

Structural basis for SHOC2 modulation of RAS signaling: Authors' response to referees. Revision of original submission: 2021-10-15837

We thank the referees for their thoughtful comments. We have taken the opportunity to perform more experiments and to clarify specific questions raised by each reviewer, and we believe that this process has genuinely improved the quality of the manuscript. Please find detailed responses to each question below with references to the relevant parts of the manuscript.

In the following, we have listed in **bold**, the entirety of the referees' comments and have provided point by point responses in normal text as appropriate throughout.

Referee #1 (Remarks to the Author):

The MRAS/SHOC2/PP1C complex is well-known for its ability to dephosphorylate the key NTpS 14-3-3 binding site on RAF proteins, thereby contributing to RAF activation. A structural understanding of the event is currently lacking and the authors set to solve this issue by elucidating the structure of the complex by cryoEM. Briefly, using recombinant proteins, they first demonstrated the requirement for MRAS GTP-loading for complex formation as assessed by size exclusion chromatography. Intriguingly, unlike reports by other groups, they also found that RAS oncoproteins (H, K and NRAS) also allowed RAS/SHOC2/PP1C complex formation in a GTP-dependent manner, albeit these proteins exhibited significantly lower binding affinities compared to MRAS. Using cryoEM, they then solved the structure of the MRAS/SHOC2/PP1C complex at an overall resolution of 3.0 Å. Preliminary structural information was also gathered on the KRAS/SHOC2/PP1C complex. Although too early to be conclusive, this latter complex seems reminiscent of the MRAS/SHOC2/PP1C complex, which would suggest similar mechanism of action across the various RAS-containing SHOC2/PP1C complexes. They next described the binding interfaces linking MRAS, SHOC2 and PP1C. This provided structural insights into the binding

requirements, the impact of GTP binding, substrate recognition, and the underlying basis of RASopathy mutations peppering the MRAS, SHOC2 and PP1C proteins. Finally, they note that the structure of the MRAS/SHOC2/PP1C complex offers novel therapeutic opportunities.

Overall, this is an exciting and well-written article that provides genuinely novel and insightful information on a key regulatory complex of the RAS/MAPK pathway. Although there is already a wealth of data on the topic, the current work provides at last a conclusive structural framework for their interpretation. That said, in my opinion, there are a few issues that need to be addressed to strengthen the data and eliminate some of the more speculative aspects of the work that currently reduce its significance.

Main points:

1- Each binding interface should be validated by site-directed mutagenesis followed by binding and functional assays. Although some is done indirectly by mentioning mutations found elsewhere (i.e. RASopathies or genetic screen), they should do it systematically for each surface. For instance, they should address the functional relevance of the VAF motif.

We thank referee 1 for this comment, and note that referees 2 and 3 asked very similar questions. We have performed additional experiments that validate the binding interfaces between proteins, and that we determine the effect of previously identified mutants within the complex:

- In Extended Data Fig. 6b (main text line 141), we generated three SHOC2 charge reversal mutations at the PP1C binding interface and performed binding experiments, validating the relevance of the SHOC2 charged patches in the SHOC2:PP1C interface for complex formation.
- In Extended Data Fig. 7b (main text line 169), we performed binding experiments with SHOC2AN, showing the significant contribution of the VAF motif in the SHOC2 N-terminus for complex formation. The N-terminal deletion here removed all the disordered residues in addition to the ordered VAF motif.
- In Fig. 4a-b (main text line 279), we generated a panel of 8 SHOC2/PP1C point mutants which have been previously phenotypically identified as dysregulating signaling. Binding experiments with these reagents showed that increased or decreased binding affinity at various binding interfaces correlated with the expected phenotypic outcome.

2- line 167: They claim that the few amino acid differences between PP1C isoforms likely do not explain their distinct biological or binding affinities to SHOC2. They should address this point directly either by testing (binding and function) relevant amino acid changes or by comparing the binding affinities of the different PP1C isoforms.

This is an interesting question, particularly given the affinity differences between RAS isoforms for complexation with SHOC2:PP1C. We have generated additional purified PP1C isoforms and performed experiments that confirm our original structure-based hypothesis (Extended Data Fig. 7c) that PP1C α , β and γ are all able to participate in the SHOC2:PP1C:RAS complex. Specifically, in Extended Data Fig. 7 (main text line 174), we performed the following experiments:

- at **d**, we validated SHOC2:PP1C:RAS complex formation with all isoforms by SEC,
- at **e**, we determined that all PP1C isoforms show increased catalytic efficiency against a BRAF NTpS substrate when complexed with SHOC2 and RAS,
- at **f**, we showed that three way complex binding affinity is similar between PP1C isoforms,
- at **g**, we showed that all PP1C isoforms are able to bind to MRAS(GDP).

3- Given the significantly higher binding affinity of MRAS compared to H,K,NRAS GTPases, compounded with the fact that other groups failed to detect significant binding of non-MRAS GTPases to SHOC2/PP1C, the relevance of H,K,NRAS forming complexes with SHOC2/PP1C in normal physiological conditions remains uncertain and needs to be addressed. A- Do any of the H,K,NRAS/SHOC2/PP1C complexes endogenously exist in cells? B- Are they genuinely involved in RAF NTpS dephosphorylation in the absence of MRAS? Otherwise, their results remain speculative.

We thank referee 1 for this question, and understand its importance, particularly in light of conflicting literature reports about the role of H/K/NRAS in this pathway. As noted in the manuscript, SHOC2:PP1C:H/K/NRAS complexes have not been successfully immunoprecipitated from cells in the literature, even under circumstances where some components are overexpressed. However, given the ~20-40-fold lower apparent affinity of H/K/NRAS for complex formation, this lack of observation might have been because H/K/NRAS based complexes do not bind strongly enough to survive the binding and

washing steps of a co-immunoprecipitation experiment, which may poorly recapitulate the native environment of the putative complex in the cellular context. In addition, intrinsic RAS GTP hydrolysis during the course of the experiment under normal physiological conditions with wild-type RAS would result in RAS(GDP), which would further destabilize the complex. In our case, we are able to generate stable complexes by using higher concentrations of recombinantly purified protein with a non-hydrolysable analog of GTP. Given this background, we considered it was unlikely that observing a native H/K/NRAS containing complex from cells would be possible using classical immunoprecipitation methods.

We turned to functional cell based data to determine whether our observed *in vitro* H/K/NRAS binding data is reflected in real biological effects. Indeed, our analysis of the Cancer Dependency Map data corresponding to an exhaustive list of 1061 tumor cell lines in Fig. 3c-d (main text line 242) contains two strong implications about the relevance of H/K/NRAS in different contexts:

- In tumor cell lines strongly dependent on SHOC2 for growth, MRAS knockout is *always* well tolerated, suggesting that other RAS isoforms are able to compensate and mediate activity through the SHOC2 axis.
- Where tumor cell lines have strong H/K/NRAS driver mutations and are strongly dependent on SHOC2 for growth, H/K/NRAS knockout is poorly tolerated, and this is significantly correlated to the cells' dependence on SHOC2. This suggests that in an oncogenic setting where high levels of GTP loaded H/K/NRAS are present, these RAS isoforms become the dominant drivers of SHOC2 dependent activity.

We understand that we have made these conclusions based on tumor cell lines, whereas referee 1 asks about H/K/NRAS in a normal physiological setting. Whether H/K/NRAS play the same role in a non-tumor setting is still an open question which we believe would be very difficult to answer conclusively for the following reasons. As outlined above, detection of a direct interaction between SHOC2 and H/K/NRAS could be impaired due to weaker affinities in a context outside the cell. It would also be difficult to use a similar genetic approach to answer the question in a wild type setting, since SHOC2 or MRAS knockout produces such a mild phenotype. Based on all of our other results, it seems very likely that SHOC2:PP1C:H/K/NRAS can form in normal wild type cells. But whether these complexes are present at high enough levels to be biologically relevant in this setting would depend on the fine balance of different RAS isoform expression levels and GTP loading states of each isoform within the cell. We appreciate that although this is a strong implication for normal cells, it could be considered speculative, and so we have restricted our claims to the cancer cell setting where the data is clear.

4- PP2A has also been reported as a phosphatase involved in dephosphorylating the NTPS sites in RAF proteins following RAS activation (Ory et al. Curr Biol 2003). Is it possible that only MRAS forms physiological complexes with SHOC2/PP1C, whereas H,K,NRAS use other mechanisms, such as PP2A-dependent mechanisms?

A structural analysis of PP2A suggests that it would not form a complex with SHOC2:RAS. An overlay of PP1C and PP2A shows that key SHOC2 interacting charged patches on PP1C do not exist on PP2A (Reviewer Response Fig. 1). On the opposite side of the phosphatases, representing the PP1C:RAS interaction surface, PP1C/PP2A electrostatics are also different, suggesting that PP2A would not be able to compensate for PP1C in the same way.

Reviewer Response Fig. 1 | Electrostatic views of paired perspectives of PP1C (left) and PP2A (right), highlighting SHOC2 binding sites (top) and RAS binding site (bottom).

PP2A is a relevant regulatory phosphatase, but it seems unlikely this is through a mechanism similar to that of SHOC2:PP1C:RAS. We have not included this analysis in the manuscript because we cannot formally rule out that such a complex may form, and we did not want to distract from the main PP1C focused message.

5- Since RAS apparently uses overlapping interfaces for binding RAF and SHOC2/PP1C, an interesting possibility is that two distinct RAS molecules are required for driving RAF activation, namely, one that recruits RAF through RBD binding, and one that recruits SHOC2/PP1C to dephosphorylate RAF's NTpS site. This insight is conceptually novel and likely applicable to other small GTPases involved in other complexes/signaling mechanisms. Unfortunately, the concept remains speculative at this point. A formal demonstration would strongly strengthen the reach of their work.

We respectfully disagree that the concept of requirement of two RAS molecules is speculative, based on our data. We consider that our existing data, combined with the literature, proves the point that two RAS molecules are required for substrate dephosphorylation, with reasoning as follows. We have firstly shown that SHOC2:PP1C and RAF cannot bind the *same* RAS:

- Our SHOC2:PP1C:RAS structure, in conjunction with the literature, shows there is a near-complete overlap in the RAS binding interface for SHOC2:PP1C and RAF RBD-CRD at Fig. 2d (main text line 197).

- Our solution based binding data further demonstrate the mutual incompatibility of these interactions at Extended Data Fig. 8a (main text line 202).

We can also conclude from our data and the literature that SHOC2:PP1C and RAF *must* be bound to two molecules of RAS in the cell for efficient dephosphorylation:

- RAS is the only membrane anchored molecule in the SHOC2:PP1C:RAS complex (Fig. 1d, main text line 112, Extended Data Fig. 4d), and the only membrane anchored molecule in the RAS:RAF complex. To bring together both of these components on the membrane therefore requires one RAS to be bound to each.
- The only point of interaction we identified between the SHOC2:PP1C:RAS complex and the RAS:RAF complex was the RAF NTpS binding in the PP1C active site. This is a low affinity interaction (Fig. 4c, Extended Data Fig. 8a, 9c, main text line 319).
- BRAF NTpS dephosphorylation is strongly correlated with RAS activation (GTP loading state). Any PP1C and SHOC2:PP1C which may be free in the cytosol do not significantly contribute to RAS dephosphorylation without being bound to RAS (Rodriguez-Viciano et al., *Mol Cell* (2006)).

As a formal demonstration of this requirement, we consider that we would need to simultaneously show the following:

- (a) A co-localization between KRAS (bound to RAF:RBD-CRD) and MRAS (bound to SHOC2:PP1C);
- (b) In a membrane environment;
- (c) While at the same time show that such co-localization directly results in a concomitant dephosphorylation of the RAF NTpS.

We consider that fulfilling all 3 of the above *in vitro* using recombinant proteins (farnesylated, palmitoylated, proteolyzed and methylated RASs) or in cells would be significantly time consuming and technically challenging. Even if successful, it is unclear that such a result would advance our understanding of the system beyond what we have described in our manuscript already, based on our structural and biochemical analyses and the existing literature.

5b - As a corollary to this point, could H,K,NRAS work in concert with MRAS for optimal RAF activation? For example, one class of GTPases (H,K,NRAS) might serve to recruit RAF, while the other (MRAS) serves to recruit SHOC2/PP1C?

This is an interesting proposition, which we think is supported by the data. Considering all together the data we had presented in our first submission, new cell based data analysis in our revised manuscript, as well as more literature precedent, we are able to state this concept more explicitly than we did in our original manuscript (main text line 385).

From a biochemical standpoint, we have shown that there is a ~200 fold binding affinity “selectivity” advantage for MRAS against SHOC2:PP1C vs RAS RBD, when compared to H/K/NRAS (main text line 387). From a cellular standpoint, MRAS appears to be the dominant RAS for SHOC2 mediated signaling outside of the mutant H/K/NRAS driven cancer setting (see also referee 1, response 3). Together, this data suggests that under normal physiological conditions, MRAS is the primary isoform involved in

SHOC2:PP1C binding, and H/K/NRAS are the primary isoforms involved in RAF RBD-CRD binding. Interestingly, the cellular data suggests that different expression and GTP loading levels of the different RAS isoforms can overcome this ~200x selectivity advantage, so this general proposition about isoform selectivity does not always hold true (e.g. in the case of cells containing H/K/NRAS driver mutations).

Referee #2 (Remarks to the Author):

In this manuscript Liau et al., present a cryo-EM structure of the SHOC2-PP1C-RAS complex, as well as additional structural and biochemical data on the regulation of BRAF dimerization and activation by the SHOC2:PP1C:RAS complex. The reported structures, accompanied by the Size Exclusion Chromatography (SEC) and Surface Plasmon Resonance (SPR) data support a model by which a) formation of the complex requires GTP-bound RAS, b) MRAS has higher affinity for binding in the complex, compared to K/N/HRAS and c) even though a peptide containing S365 readily binds and is dephosphorylated by the complex, ordered domains of BRAF are not able to bind it. The data indicate that two different RAS molecules are required for BRAF activation in the membrane, one that binds the RBD domain of BRAF and the other that participates in the SHOC2:PP1:RAS complex. The data are convincing and consistent with a large body of previous literature (Rodriguez-Viciana et al, Mol Cell, 2006, as well as more

recent studies on the topic by Young et al., PNAS 2018, Boned del Rio et al PNAS, 2019), to which they provide additional structural and biochemical support. An important question that remains unresolved in the field is to define the contribution of mutationally activated H/N/KRAS in relation to MRAS in regulating BRAF S365 dephosphorylation in cells. To what extent oncogenic K/N/HRAS mutations are sufficient to bypass the need for MRAS or SHOC2 activity? Answering these questions may have important translational implications in terms of identifying potentially new therapeutic approaches to target MAPK signaling. These questions are only indirectly addressed in the manuscript, without cell-based experimental evidence.

Further, the manuscript would be strengthened with experimental validation of some critical predictions: The authors prepared chimeric KRAS with MRAS N and or C termini and did not observe increased binding to the SHOC2:PP1C compared to KRAS. They could prepare based on the structure, MRAS in which critical amino acids are replaced with the analogous of KRAS, and show weaker binding to the complex. Similarly, in Fig 4, reported loss of function and gain of function mutations in SHOC2, PP1C and MRAS are only explained using the generated structures, but not experimentally validated in the biochemical assays the authors have in hand.

Other comments:

- Fig 1b is the only main fig. panel that presents the cryo-EM density throughout the paper (together with the local res representation in ext fig. 3d). Additional representations would improve the presentation of the data. For example the reconstruction can also be shown in higher threshold so that structural elements can be more readily recognized. Moreover, a figure showing fit of the atomic model to the cryo-EM density, with representative structural elements shown in closeup, is needed.

- An angular distribution plot is not presented for the data and it is needed given that issues of preferred orientation for some of the complexes are discussed.

- In ext Fig. 3a, the “representative” micrograph shown exhibits severe issues with particle distribution. The rationale for presenting this micrograph is unclear as it generally would be considered very poor. Were all micrographs in the dataset of similar appearance or could a “better” micrograph be presented? In addition, an inset in higher magnification is necessary to present a better view of the particles in the micrograph.

- The 2D class averages presented need to be increased in size, as their small size significantly hinders a proper evaluation of their quality.

- Was any masking used with local refinement or 3D classification to try to improve the parts of the protein that seem to be more flexible (lower resolution towards the ends in the local res map)?

Overall, although the main conclusions of the manuscript are not entirely surprising, given the current biochemical and cell-based conceptual understanding on how the SHOC2-PPI1C-MRAS complex dephosphorylates and stabilizes dimeric RAF, the study provides the first cryo-EM structure of a complex critical for MAPK activation, as well as useful structural and biochemical information.

Point by Point responses:

An important question that remains unresolved in the field is to define the contribution of mutationally activated H/N/KRAS in relation to MRAS in regulating BRAF S365 dephosphorylation in cells. To what extent oncogenic K/N/HRAS mutations are sufficient to bypass the need for MRAS or SHOC2 activity? Answering these questions may have important translational implications in terms of identifying potentially new therapeutic approaches to target MAPK signaling. These questions are only indirectly addressed in the manuscript, without cell-based experimental evidence.

We thank referee 2 for this question, and understand its importance, particularly in light of conflicting literature reports about the role of H/K/NRAS in this pathway. We analyzed functional cell based data to determine whether our observed *in vitro* H/K/NRAS binding data is reflected in real biological effects. Indeed, our analysis of the Cancer Dependency Map data corresponding to an exhaustive list of 1061 tumor cell lines in Fig. 3c-d (main text line 242) contains two strong implications about the relevance of H/K/NRAS in different contexts:

- In tumor cell lines strongly dependent on SHOC2 for growth, MRAS knockout is always well tolerated, suggesting that other RAS isoforms are able to compensate and mediate activity through the SHOC2 axis.
- Where tumor cell lines have strong H/K/NRAS driver mutations and are strongly dependent on SHOC2 for growth, H/K/NRAS knockout is poorly tolerated, and is significantly correlated to the cells' dependence on SHOC2. This suggests that in an oncogenic setting where high levels of GTP loaded H/K/NRAS are present, these RAS isoforms become the dominant drivers of SHOC2 based signaling.

Further, the manuscript would be strengthened with experimental validation of some critical predictions: The authors prepared chimeric KRAS with MRAS N and or C termini and did not observe increased binding to the SHOC2:PP1C compared to KRAS. They could prepare based on the structure, MRAS in which critical amino acids are replaced with the analogous of KRAS, and show weaker binding to the complex.

We thank referee 2 for this experimental suggestion, which we have executed. We generated MRAS-KRAS chimeras with residue substitutions at the RAS:PP1C interface, and tested their complex binding affinity by TR-FRET. In Extended Data Fig. 8e (main text line 233), we found that, as predicted by our structure, replacement PP1C contacting residues on MRAS to those found on KRAS reduces the binding affinity of MRAS back to that of KRAS, highlighting the importance of this contact region for the difference in affinity between MRAS and KRAS for the ternary complex formation with SHOC2:PP1C.

Similarly, in Fig 4, reported loss of function and gain of function mutations in SHOC2, PP1C and MRAS are only explained using the generated structures, but not experimentally validated in the biochemical assays the authors have in hand.

We thank referee 2 for this comment, and note that referees 1 and 3 asked very similar questions. We have performed additional experiments that validate the binding interfaces between proteins, and that determine the effect of previously identified mutants within the complex.

- In Extended Data Fig. 6b (main text line 141), we generated 3 SHOC2 charge reversal mutations at the PP1C binding interface and performed binding experiments, validating the relevance of the SHOC2 charged patches in the SHOC2:PP1C interface for complex formation.
- In Extended Data Fig. 7b (main text line 169), we performed binding experiments with SHOC2 Δ N, showing the significant contribution of the VAF motif in the SHOC2 N-terminus for complex formation. The N-terminal deletion here removed all the disordered residues in addition to the ordered VAF motif.
- In Fig. 4a-b (main text line 279), we generated a panel of 8 SHOC2/PP1C point mutants which have been previously phenotypically identified as dysregulating signaling. Binding experiments with these reagents showed that increased or decreased binding affinity at various binding interfaces correlated with the expected phenotypic outcome.

Other comments:

- Fig 1b is the only main fig. panel that presents the cryo-EM density throughout the paper (together with the local res representation in ext fig. 3d). Additional representations would improve the presentation of the data. For example the reconstruction can also be shown in higher threshold so that structural elements can be more readily recognized. Moreover, a figure showing fit of the atomic model to the cryo-EM density, with representative structural elements shown in closeup, is needed.

We agree that these are helpful suggestions for reader understanding and have incorporated them in Extended Data Fig. 4a-b. We showcase Switch II region in MRAS since it is one of the more flexible parts of GTPases, and adopts a specific conformation in complex with SHOC2. For another example, please see Extended Data Fig. 6d, showing the fit of the atomic model to the cryoEM "VAF" sequence from SHOC2 bound to PP1C.

- An angular distribution plot is not presented for the data and it is needed given that issues of preferred orientation for some of the complexes are discussed.

We agree that this is an important piece of data, given preferred orientation issues. We have incorporated this data into Extended Data Fig. 3d.

- In ext Fig. 3a, the “representative” micrograph shown exhibits severe issues with particle distribution. The rationale for presenting this micrograph is unclear as it generally would be considered very poor. Were all micrographs in the dataset of similar appearance or could a “better” micrograph be presented? In addition, an inset in higher magnification is necessary to present a better view of the particles in the micrograph.

We chose this micrograph because it emphasizes the role of graphene oxide (GO) in this dataset. The perceived poor particle distribution is actually a flake of GO which occupies only the right hand side of the micrograph. Particle density on the GO is high because of adsorption (which is the intended effect of using GO), while it is much lower in the ice without GO.

We have not included a magnified inset, but instead used the space to increase the size of the 2D class averages (see below), which allow a more informative assessment of particle quality.

- The 2D class averages presented need to be increased in size, as their small size significantly hinders a proper evaluation of their quality.

We thank referee 2 for this suggestion, which we have implemented at Extended Data Fig. 3b.

- Was any masking used with local refinement or 3D classification to try to improve the parts of the protein that seem to be more flexible (lower resolution towards the ends in the local res map)?

We did not mask out any regions of the protein to arrive at the final structure. Masking was one approach we tried during processing, where we masked out the flexible SHOC2 C-terminus in an attempt to improve the resolution of the remainder of the complex. However, this did not result in an improvement in resolution in practice. We did not try the inverse process to this (i.e. including only the SHOC2 C-terminus), as this would not have left enough mass to align particles.

A likely explanation for masking being ineffective is that the 126kDa protein complex is relatively small for cryoEM, with only ~110kDa of this visible in the final map. Any gains in resolution which might have been gained from removing a less-ordered portion of the complex seemed to be more than offset by the reduction in total mass available for particle alignment during refinement. Additionally, our final structure used a “non-uniform” refinement approach (Punjani et al., *Nature Methods* (2020)). This spatially varies the Fourier Shell Correlation (FSC) cutoff during refinement according to local resolution (as opposed to using a single, global, FSC cutoff value). Especially for small and membrane proteins, this method is able to account well for local resolution variation across a structure. Indeed, we found that in this case, non-uniform refinement performed significantly better than uniform (traditional) refinement with or without masking. Overall, the C-terminus of SHOC2 was still resolved more poorly than the rest of the map, because it seems that it genuinely is a flexible region of the protein, likely important for substrate recognition (See Fig. 4g, Extended Data Movies 1-5).

Referee #3 (Remarks to the Author):

The RAS-RAF axis is critical for many biological/pathological processes, in particular tumorigenesis. This work by Liau et al. demonstrates RAS isoform specificity and its GTP dependence for formation of the RAS-SHOC2-PP1 complex, and reports a high-resolution cryo-EM structure of this important complex. Main biochemical and structural results in this manuscript appear novel, technically solid and interesting. However, more biochemical studies and maybe more structural characterization are needed to clarify some important aspect of this work.

Main issues:

1. The most surprising aspect of the current RAS-SHOC2-PP1 complex structure is that RAS SwI and SwII are directly involved in its interaction with SHOC2/PP1, and thus RAF cannot be directly recruited to the RAS-SHOC2-PP1 complex for dephosphorylation by PP1. Interestingly, while no apparent binding could be detected, the RAS-SHOC2-PP1 complex has a significantly (~18X) lower K_m for the BRAF pS365 peptide than PP1 per se. How is the specificity and lower K_m for the RAF pS365-peptide achieved at the PP1 active site in the complex? How about other RAF phospho-peptides (e.g. pS259 RAF1, pS214 ARAF)? I wonder if authors have tried to visualize how the RAF pS365-peptide can be recognized by the phosphatase active site of the RAS-SHOC2-PP1 holoenzyme. One of possible ways to stabilize the transient BRAF pS365-peptide substrate binding for cryo-EM analysis is to fuse the RAF phospho-peptide covalently to the C-terminus of PP1C using the intein protein fusion approach. If cryo-EM analysis of a RAF phospho-peptide containing complex is not possible, a molecular dynamics simulation of the RAF phospho-peptide recognition by this holoenzyme complex, based on current cryo-EM structure, is needed to obtain a somewhat clear picture on how the MRAS-SHOC2-PP1 complex promotes RAF dephosphorylation and activation. Are RAS and SHOC2 directly involved in RAF substrate recognition or merely provide a membrane anchor for PP1? This issue is not only critical for mechanistic understanding but also for developing novel RAS/RAF inhibitors targeting at the RAF recognition site.

We thank referee 3 for this insightful proposal, which led us to perform a series of experiments that have greatly increased our understanding of SHOC2:PP1C:RAS substrate recognition. We firstly performed the PP1C vs SHOC2:PP1C:RAS peptide dephosphorylation experiment on a larger panel of substrate

peptides, as suggested, as well as negative control peptides (Fig. 4c, main text line 306). This result allowed us to extract information about the sequence determinants of specificity.

These determinants of specificity, along with sequence conservation and chemical properties of the SHOC2:PP1C:MRAS surface, allowed us to create a model which we validated by molecular dynamics (MD) simulations, as suggested. This has allowed us to show that the SHOC2 C-terminal hydrophobic patch is a key interacting region, driving specificity for the NTPase, as explained in Fig. 4d-g (main text line 325). The ensemble of structures generated by MD is not as precise as a single X-ray or cryoEM structure, but this is likely because the substrate binding in this case is in an ensemble of closely related states and means that there is no single structure which precisely describes the binding mode.

Our data also addresses the specific question, “are RAS and SHOC2 directly involved in RAF substrate recognition or merely provide a membrane anchor for PP1?” Our kinetic data show that NTpS peptide

must be interacting with proteins outside of PP1C in the context of the complex. Extended Data Fig. 8b shows the effect of SHOC2 and KRAS addition at varying concentrations of each protein, with increasing activity as more SHOC2 is added. (The primary effect of adding more KRAS appears to be indirect, as a result of increasing the bound fraction of SHOC2:PP1C.) These experiments were performed with recombinant proteins in solution, with no membrane present. Only two physical phenomena can explain the increase in catalytic activity of PP1C: either 1) the peptide is interacting directly with regions outside of PP1C, or 2) complex formation is allosterically altering PP1C to increase its activity. We can rule out option 2, because increased catalytic activity is not observed against the Non-Target Peptides (Extended Data Fig. 9a), nor against generic substrate PNPP (Extended Data Fig. 9b).

We also performed a cryoEM experiment as per referee 3's suggestion. Consistent with the above interpretation of the MD results, we did not obtain a definitive peptide bound structure, and so have not included this data in the manuscript. We provide here a brief outline of this experiment. We generated MRAS:BRAF-NTpS fusion proteins with the BRAF NTpS attached to MRAS N-terminus by a flexible GGSG linker that is long enough to reach across the PP1C surface. One construct contained a S-E (pS charge mimetic mutation) at the pS site, and the other was wild type. We assembled the SHOC2:PP1C:MRAS:BRAF NTpS complex with the S-E mutant and determined its cryoEM structure. We did not see any density for the peptide in the PP1C active site (Reviewer Response Fig. 2), nor any obvious patches of unexplained density elsewhere on the complex. Mass spectrometry clearly showed that the peptide fusion was successful, so the most likely explanation for the cryoEM result is that the peptide has a low residence time on the complex.

Reviewer Response Fig. 2 | SHOC2:PP1C:MRAS:BRaf NTPs(S-E) complex models representative pS peptide (pS +/- 2 residues) bound to the PP1C active site, inserted into, **a**, sharpened map contoured to 0.15, **b**, unsharpened map contoured to 0.1, **c**, sharpened map contoured to 0.17, **d**, unsharpened map contoured to 0.13. Structure was solved using data collected on a Glacios microscope with fewer particles than the structure reported in the manuscript, so map quality is not equivalent. Sharpening B factor = -195 Å².

We were also able to specifically phosphorylate the NTPs of the MRAS:BRaf NTPs wild type construct using the kinase AKT1, the known kinase for RAF NTPs, and used it to assemble a SHOC2:PP1C:MRAS:BRaf-NTPs complex, with the use of “phosphatase dead” PP1C H125A mutant. We found that even PP1C H125A had a residual level of catalytic activity. This complex therefore required fast assembly just prior to grid freezing, without our normal SEC purification, to avoid dephosphorylation of the NTPs. Likely because of this different protocol, we did not observe any 2D complex particles, and were unable to solve its structure.

Lastly, we also generated BRAF NTpS peptide fused to the PP1C C-terminus (closest to the suggestion made by the referee), but PP1C was rendered highly unstable and insoluble by this fusion, which prevented complex assembly. Regardless, our cryoEM data with BRAF NTpS fusion at the MRAS N-terminus above reflects that the peptide would have had a low residence time, even if we could assemble this complex.

Given the high K_m values of our peptide dephosphorylation assay (Extended Data Fig. 9a), it is perhaps unsurprising that we did not observe a stable conformation of the peptide in the cryoEM structure we did obtain. This makes sense biologically: if the peptide binds too strongly to the complex, it would not unbind quickly after dephosphorylation for the next pS containing substrate to bind the enzyme. This is also consistent with our enzymatic turnover values of around 40 s^{-1} under saturating peptide conditions, suggesting that peptide unbinding from the complex is a frequent event. Therefore, the fact that we did not obtain a cryoEM structure of a stable peptide interaction does not invalidate the concept that the peptide interacts with SHOC2, rather, it is consistent with this being a weak and transient interaction.

2. Why was the dimer of MRAS-SHOC2-PP1 trimers not further structurally characterized? Could this (MRAS-SHOC2-PP1)₂ hexamer be biologically important? It is hard to see any 3D structural features of this interesting hexamer in the Extended Fig 5f. Could this low-resolution hexameric model be validated by biochemical/biophysical analysis (e.g. cryo-EM, cross-link MS)? Analysis of cross-validated low-resolution hexameric model can be very insightful.

After significant effort, we were unable to solve a high-resolution structure of the hexamer from our cryoEM dataset, despite there being a significant fraction of hexameric particles (~30%). We hypothesized in the manuscript that this was likely due to issues of preferred orientation and/or flexibility (main text line 127). It is difficult to observe 3D features of the hexamer in Extended Data Fig. 5f, because this is a model based on SAXS data, which is an inherently low resolution way of modeling proteins. The 3D models placed inside the SAXS shape envelope are to illustrate that the envelope is the right size and shape to contain the putative SHOC2:PP1C:MRAS hexamer, rather than as a specific claim about the exact structure of the hexamer.

Because of this, we mentioned the hexamer in our manuscript for the sake of completeness for readers paying close attention to the cryoEM 2D class averages, and the slight differences in elution volume on SEC for MRAS vs H/K/NRAS based complexes.

Interest in the hexamer is understandable, especially given that it might shed light on the elusive “RAS dimer” which has been controversially proposed in the literature. However, this remains an area of future interest that we consider is a very different story to that presented in this manuscript.

3. Due to flexibility of the SHOC2 scaffold, authors cannot conclude that clinical KRAS inhibitors could lead to steric clashes and thus inhibit/modulate RAS's interactions with the SHOC2-PP1 complex and subsequent RAF dephosphorylation (Fig 4b). Experimental data must be provided to support the claims. Alternatively, the last section of Results (lines 301-312) can be removed.

Please see our response below to referee 3, minor point 7. In any case, we have removed this section to provide more space to present our new Molecular Dynamics and cell based data, and because we felt it was similar to the existing GDP/GTP bound RAS comparisons in Fig. 2b.

4. Previous works have identified a number of gain- and loss-of-function mutants, which are associated with RASopathy phenotypes and can be mapped to the RAS-SHOC2-PP1 interfaces. Can some of the key mutants experimentally tested to validate structural observations?

We thank referee 3 for this comment, and note that referees 1 and 2 asked very similar questions. We have performed additional experiments that validate the binding interfaces between proteins, and that determine the effect of previously identified mutants within the complex.

- In Extended Data Fig. 6b (main text line 141), we generated 3 SHOC2 charge reversal mutations at the PP1C binding interface and performed binding experiments, validating the relevance of the SHOC2 charged patches in the SHOC2:PP1C interface for complex formation.
- In Extended Data Fig. 7b (main text line 169), we performed binding experiments with SHOC2 Δ N, showing the significant contribution of the VAF motif in the SHOC2 N-terminus for complex formation. The N-terminal deletion here removed all the disordered residues in addition to the ordered VAF motif.
- In Fig. 4a-b (main text line 279), we generated a panel of 8 SHOC2/PP1C point mutants which have been previously phenotypically identified as dysregulating signaling. Binding experiments with these reagents showed that increased or decreased binding affinity at various binding interfaces correlated with the expected phenotypic outcome.

Minor points:

1. Is the described flexibility of SHOC2 scaffold a result of collective flexibility cross the entire LRR structure or a hinge-like motion? It has been previously predicted that SHOC2 contains a hinge in the middle of its LRR fold (Kwon et al, 2021, MCB). A more detailed structural comparison between the RAS-SHOC2-PP1 cryo-EM structure and the SHOC2 crystal structure may help.

The flexibility of SHOC2 is collective change in pitch along the whole LRR. Our X-ray and cryoEM structures show that the “hinge” region proposed in some literature does not exist, and appears to be an erroneous computational structure prediction (e.g. Kwon et al., MCB (2021) Fig. 1C).

We have slightly updated Extended Data Fig. 4f, and now measure the angle of “twisting” of SHOC2 from the N terminal LRR. We realize that our previous depiction measured from the 8th LRR may have left the impression that this LRR was somehow “special” or represented the proposed hinge region. The 9 degree twist we have measured is instead intended to demonstrate the cumulative conformational change over all of the SHOC2 LRRs.

2. There is a lack of description, in fig legends or methods, on how 3D structural alignment or superposition is done. Were all shown results from globally (or subunit/region-guided) superposition?

We thank referee 3 for pointing out this possible point of confusion. We have updated the relevant figure legends to point out more explicitly where non-standard alignments have been used. Where not noted specifically, we understand a global alignment to be an implied standard practice (e.g. PP1C:PP1C and RAS:RAS alignments in Extended Data Fig. 4c).

3. Can MRAS(GDP) bind to other PP1C isoforms?

We thank referee 3 for this question, which we have shown is the case by performing additional experiments. We have generated additional purified PP1C isoforms, and in Extended Data Fig. 7g (main text line 174), we show that all PP1C isoforms are able to bind to MRAS(GDP). We also note a series of other experiments in Extended Data Fig. 6g that further emphasize the biochemical similarities between PP1C isoforms:

- at **d**, we validated SHOC2:PP1C:RAS complex formation with all isoforms by SEC,
- at **e**, we determined that all PP1C isoforms show increased catalytic efficiency against a BRAF NTPS substrate when complexed with SHOC2 and RAS,
- at **f**, we showed that three way complex binding affinity is similar between PP1C isoforms.

0. Use of GCP and GNP nomenclature should be unified. GMP-PNP was also used in

this ms. Thank you for this suggestion, we have unified the nomenclature to GCP and GNP.

1. In Fig 1, it would be useful to highlight the MRAS SwI and SwII (e.g. in red) in the overall structure

We have not implemented this change to keep Fig. 1c as uncluttered as possible. Fig. 1 is intended to give a broad overview of all three proteins at the same time. SwI and SwII are hidden at the back of the structure in this view, so highlighting them makes the figure more confusing. We focus on SwI and SwII specifically in Fig. 2b-c.

2. Comparison of areas of buried surfaces for each interface?

We go into significant detail about each binding interface in the manuscript. We did not want to provide a buried surface area number for each interface, as we felt some readers might latch onto a comparison of these values as some kind of universal metric as to the importance of each interface.

3. As the SHOC2 scaffold has some structural flexibility as shown by authors, 'steric clash' (line 180) is not a good explanation for a lack of binding of the GDP-bound Ras.

We respectfully disagree with this comment, which we believe overestimates the degree of flexibility we have observed in SHOC2. PP1C and RAS bind towards the N terminal end of SHOC2. The relative positions of SHOC2 and RAS are constrained by the need to ensure that PP1C can contact both of them. An alignment of our cryoEM or X-ray structures of SHOC2 based on the first four LRRs shows that RAS:GDP would clash with either (Reviewer Response Fig. 3).

Additionally, although the 9 degree twisting we observe between both SHOC2 structures is suggestive of some level of flexibility, this level of movement is not so great that it would avoid the steric clash from RAS:GDP.

Reviewer Response Fig. 3 | SHOC2:PP1C:MRAS(GCP). MRAS(GCP) (salmon) and PP1C (purple) bound to SHOC2 from cryoEM (green), aligned with SHOC2 from X-ray (orange) based on the N terminal capping helix and first four LRRs.

4. In many cases, including Fig 1 and 4c, molecular labels can be color-coded – same as the colors of corresponding structures.

Thank you for this excellent suggestion, which we have implemented throughout the manuscript where appropriate.

5. Crystallographic statistics for the highest resolution data bin do not look good (Table S1). The resolution should be cut to a lower resolution (3.3 or 3.4?).

We used a CC1/2 based resolution cutoff for this dataset, as is preferred practice in contemporary crystallography (Karplus and Diederichs, *Current Opinion in Structural Biology* (2015)), with our chosen cutoff of ~30% being well within reason for this metric. While this results in R_{sym} and I/σ statistics which would have classically been considered unacceptable, such an approach nonetheless resulted in improved map quality.

0. How does SCRIB play a regulatory role for the RAS-SHOC2-PP1 complex?

We accept that SCRIB is an important regulator, but our current data do not shed any light on its role. We can speculate that SCRIB might, for example, be involved in interactions with some of the RASopathy causing mutations which we found do not affect SHOC2:PP1C:RAS complex formation (main text line 292).

Reviewer Reports on the First Revision:

Referees' comments:

Referee #1 (Remarks to the Author):

The authors have done a great job and responded satisfactorily to most of my comments. The only point that remains for me is the functional relevance of the VAF motif (point #1 in my initial comments). In response, the authors showed that an N-terminally truncated version of SHOC2, lacking the first 90 or so amino acids (SHOC2 Δ N), had a greatly reduced ability to support complex formation. Since the VAF motif is part of these sequences, they concluded that this shows its significant contribution to complex formation. I don't think this evidence supports the importance of this motif. They would need to mutagenize it specifically. A single point mutation should suffice, as they did for other loss of function mutations that they tested in their study. Alternatively, they should tone down their claim (line 171) and mention that their results suggest (as opposed to indicate) that the VAF motif in SHOC2 is functionally relevant.

Referee #2 (Remarks to the Author):

The authors successfully addressed my concerns and the manuscript is now acceptable for publication.

Referee #3 (Remarks to the Author):

In this revised version, authors have added important additional data, including biochemical analysis of previously identified RASopathia mutants within the complex, and how pRAF substrate peptides might be recognized by the complex. These new data, along with improved figures and discussions, have addressed all my main concerns. I think it is a highly significant piece of work.

Author Rebuttals to First Revision:

Structural basis for SHOC2 modulation of RAS signaling: Authors' response to referees.

2022.04.14

Revision of first re-submission

We thank the referees for their review of our revised manuscript and for agreeing that our manuscript is acceptable for publication in Nature.

Please see below for the Referee's comments and our response to the only question remaining from Reviewer 1.

Referee #1 (Remarks to the Author):

The authors have done a great job and responded satisfactorily to most of my comments. The only point that remains for me is the functional relevance of the VAF motif (point #1 in my initial comments). In response, the authors showed that an N-terminally truncated version of SHOC2, lacking the first 90 or so amino acids (SHOC2 Δ N), had a greatly reduced ability to support complex formation. Since the VAF motif is part of these sequences, they concluded that this shows its significant contribution to complex formation. I don't think this evidence supports the importance of this motif. They would need to mutagenize it specifically. A single point mutation should suffice, as they did for other loss of function mutations that they tested in their study. Alternatively, they should tone down their claim (line 171) and mention that their results suggest (as opposed to indicate) that the VAF motif in SHOC2 is functionally relevant.

We thank the reviewer for their comments on our manuscript and accepting our manuscript in principle. We understand the point the reviewer is making. In response, as suggested by the reviewer, we will choose to tone down the claim, and have changed the text to reflect that our results “suggest” as opposed to “indicate” that the VAF motif is functionally relevant.

Referee #2 (Remarks to the Author):

The authors successfully addressed my concerns and the manuscript is now acceptable for publication.

We thank referee #2 for their acceptance of our manuscript

Referee #3 (Remarks to the Author):

In this revised version, authors have added important additional data, including biochemical analysis of previously identified RASopathia mutants within the complex, and how pRAF substrate peptides might be recognized by the complex. These new data, along with improved figures and discussions, have addressed all my main concerns. I think it is a highly significant piece of work.

We thank referee #3 for saying that our manuscript is highly significant work. We completely agree. We also thank the reviewer for their acceptance of our manuscript for publication.